# Genomic programming of IRF4-expressing human Langerhans cells

Sofia Sirvent[1], Andres F. Vallejo [1], James Davies[1], Kalum Clayton [1], Zhiguo Wu[2], Jeongmin Woo[3], Jeremy Riddell[4], Virendra K. Chaudhri [2,11], Patrick Stumpf [5], Liliya Angelova Nazlamova[1], Gabrielle Wheway [5], Matthew Rose-Zerilli[6], Jonathan West [6,7], Mario Pujato[8], Xiaoting Chen [4], Christopher H. Woelk [9], Ben MacArthur [6,7], Michael Ardern-Jones [1], Peter S. Friedmann[1], Matthew T. Weirauch [4,10], Harinder Singh [2,10,11]* & Marta E. Polak [1,7]*

Langerhans cells (LC) can prime tolerogenic as well as immunogenic responses in skin, but the genomic states and transcription factors (TF) regulating these context-specific responses are unclear. Bulk and single-cell transcriptional profiling demonstrates that human migratory LCs are robustly programmed for MHC-I and MHC-II antigen presentation. Chromatin analysis reveals enrichment of ETS-IRF and AP1-IRF composite regulatory elements in antigen-presentation genes, coinciding with expression of the TFs, PU.1, IRF4 and BATF3 but not IRF8. Migration of LCs from the epidermis is accompanied by upregulation of IRF4, antigen processing components and co-stimulatory molecules. TNF stimulation augments LC cross-presentation while attenuating IRF4 expression. CRISPR-mediated editing reveals IRF4 to positively regulate the LC activation programme, but repress NF2EL2 and NF-kB pathway genes that promote responsiveness to oxidative stress and inflammatory cytokines. Thus, IRF4-dependent genomic programming of human migratory LCs appears to enable LC maturation while attenuating excessive inflammatory and immunogenic responses in the epidermis.

[1] Clinical and Experimental Sciences, Sir Henry Wellcome Laboratories, Faculty of Medicine, University of Southampton, SO16 6YD Southampton, UK. [2] Division of Immunobiology & Center for Systems Immunology, Cincinnati Children's Hospital Medical Center, Cincinnati, OH 45229, USA. [3] Samsung Genome Institute, Samsung Medical Center, Seoul, South Korea. [4] Center for Autoimmune Genomics and Etiology, Cincinnati Children's Hospital Medical Center, Cincinnati, OH 45229, USA. [5] Human Development and Health, Faculty of Medicine, University of Southampton, SO17 1BJ Southampton, UK. [6] Cancer Sciences, Faculty of Medicine, University of Southampton, SO16 6YD Southampton, UK. [7] Institute for Life Sciences, University of Southampton, SO17 1BJ Southampton, UK. [8] Division of Biomedical Informatics, Cincinnati Children's Hospital Medical Center, Cincinnati, OH 45229, USA. [9] Merck's Exploratory Science Center, Cambridge, MA 02141, USA. [10] Department of Pediatrics, University of Cincinnati College of Medicine, Cincinnati, Ohio 45229, USA. [11] Present address: Center for Systems Immunology, Departments of Immunology and Computational and Systems Biology, The University of Pittsburgh, Pittsburgh, PA 15213, USA. *email: harinder@pitt.edu; m.e.polak@soton.ac.uk

Langerhans cells (LC) reside in the epidermis as a dense network of immune system sentinels. They are uniquely specialised at sensing the environment by extending dendrites through intercellular tight junctions to gain access to the *stratum corneum*, the outermost part of the skin. LC are highly specialised antigen presenting cells, priming protective immune responses against pathogens encountered via the skin, such as viruses[1–3], bacteria[4] and fungi[5]. They also promote responses to chemical sensitisers[6,7]. Their position in the outermost layers of the skin barrier and their capacity to sense dangerous perturbations to their environment make them a critical first line of defence in the skin. LCs also appear to play a vital role in maintaining immune homoeostasis in the skin by activating skin-resident memory T regulatory cells[5]. In the context of foreign pathogens, LCs more effectively induce activation and proliferation of skin-resident effector memory T cells and dampen memory T regulatory cell responses. Thus, it is important to elucidate regulatory pathways and mechanisms in LCs that enable their context-dependent functions in promoting tolerogenic as well as immunogenic responses in the epidermis.

LCs are highly efficient at presenting exogenous antigens in the context of MHC Class II thereby priming antigen-specific CD4 T cells[8]. Such responses include the activation of naïve CD4 T cells in skin-draining lymph nodes as well as resident memory CD4 T cells in the skin[5]. LCs are also capable of efficient cross-presentation in which exogenous antigens are presented on MHC class I, resulting in activation and expansion of antigen-specific effector CD8 T cells[2,5,9,10]. Such cross-presentation becomes particularly important for adaptive immune responses against viruses and also cancerous cells that have evolved immune evasion mechanisms that inactivate DCs[11,12].

Activation of skin immune responses requires participation of epidermal cells in collaboration with LCs; cross-talk between the epidermal and immune components being critical. For example, in models of cutaneous viral infection, including vaccinia virus, only skin structural cells support virus replication, while immune cells, including LCs, are infected abortively[13], necessitating antigen transfer between epithelial and immune cells. Cytokine signalling from keratinocytes impacts LC development and function. Epidermal TGF-β and BMP7 are required for LC development and tolerogenic function in vitro[14–16] and cytokine signalling via thymic stromal lymphopoietin (TSLP) in atopic skin disease polarises skin dendritic cells to prime Th2 and Th22 CD4 T-cell responses, while reducing the ability of LCs to cross-prime CD8 T cells[17–19]. In contrast, TNF, a pro-inflammatory cytokine released in the skin by a variety of cell types, including keratinocytes and fibroblasts as well as dermal macrophages and neutrophils, provides a key component of cutaneous anti-viral immune responses. Numerous reports demonstrate, that keratinocyte-derived TNF delivers highly potent signals inducing LC immunogenic function and ability to present antigens[2,18,20–22] and enhances their ability to prime anti-viral adaptive immunity[23]. Cross-talk between LC and surrounding keratinocytes, coupled with their ability to cross-present antigens expressed by other cells to skin-resident and infiltrating T lymphocytes, defines the major role of LC in skin immunity and allows them to initiate efficient adaptive immune responses in the context of skin infection[5,24,25]. At the same time, continuous trafficking of cutaneous self-antigens by LCs to regional lymph nodes promotes self-tolerance[26,27]. At present, little is known about the genomic mechanisms which programme LC functions in homoeostasis and inflammation and how epidermal-derived signals modify such programming.

Like macrophages, LCs originate from yolk-sac progenitors and populate the epidermis during embryonic life. However, functionally, they are more similar to conventional DCs in their ability to efficiently present and cross-present antigens to prime T-cell responses[28,29]. Although LCs are among the most efficient antigen presenting cells, their transcriptional networks appear to be distinct from those of both macrophages and DCs[22,30,31], warranting deeper analyses. In all DC subtypes studied, two transcription factors of the interferon regulatory factor (IRF) family, IRF4 and IRF8, have emerged as key players in their development and function[32–36]. IRF4 and IRF8 control a wide range of DC functions. These include induction of innate responses elicited via pattern-recognition receptors TLR, NOD, and RIG during viral and bacterial infection[37–39], migration and cell activation, antigen uptake, presentation of MHC-I and MHC-II-restricted antigens[18,34,40], and the priming of immunogenic as well as tolerogenic CD4 T-cell responses[40,41]. Interestingly, the ability of murine DC subsets to efficiently activate CD8 and CD4 T cells, depending on the presentation of antigen in the context of MHC class I and II, is determined by the relative expression of IRF8 and IRF4, respectively[34]. Furthermore, in murine CD8α DCs cross-presentation is critically dependent on BATF3/IRF8 complexes[42–45]. Unlike the case with DCs, murine LC development does not require IRF4 or IRF8, and human LCs can develop in the absence of IRF8[30,32,46,47]. Nevertheless, the functions of IRF4 and/or IRF8 in regulating the genomic programming of migratory human epidermal LCs remain to be explored.

Here we show using transcriptional and chromatin profiling that migratory LCs are robustly programmed for MHC-I and MHC-II antigen presentation as well as mitochondrial oxidative phosphorylation. LCs express the transcription factors PU.1, IRF4 and BATF3, but not IRF8. LC migration from the epidermis enhances expression of antigen processing components, and the co-stimulatory molecules CD70, CD86 and CD40 and is accompanied by the upregulation of IRF4. TNF stimulation promotes LC cross-presentation while attenuating IRF4 expression. CRISPR-mediated editing reveals IRF4 to positively regulate the LC activation programme while repressing NF2EL2 and NFκB pathway genes that promote responsiveness to oxidative stress and inflammatory cytokines. These results suggest that IRF4-dependent genomic programming of human migratory LCs enables their maturation while attenuating excessive inflammatory and immunogenic responses in the epidermis, thereby promoting homoeostasis. Furthermore, the genomic programming of LCs is independent of IRF8 and instead utilises IRF4 in combination with PU.1 and BATF3, thereby differentiating LC from conventional dendritic cells.

## Results

**Analysing human epidermal LCs and their function.** To facilitate the analysis of the genomic programming of primary human LCs, we utilised established protocols for isolating highly pure populations of viable and functional LCs from the epidermis[2,22] (Fig. 1a–c; Supplementary Fig. 1). In agreement with our earlier findings and those of others[2,9,10], human LC after migrating from the epidermis underwent maturation and uniformly expressed high levels of CD1a, CD207 and HLA-DR (Fig. 1b; Supplementary Fig. 1a). These LCs express high levels of MHC class II as well as MHC class I complexes on their surface. Furthermore, migration from the epidermis increased LC activation status, as assessed by enhanced expression of co-stimulatory molecules: CD40, CD86 and CD70 (Fig. 1c). Such migrated LCs have been shown to efficiently present antigens to CD4 as well as CD8 T cells[2,5,9,10,18,23] (Supplementary Fig. 1b). However, the effects of migration as well as inflammatory cytokine signalling on the modulation of LC cross-presentation have not been assessed. We therefore use an established model for antigen cross-presentation[2]. Steady-state and migrated LCs were pulsed with

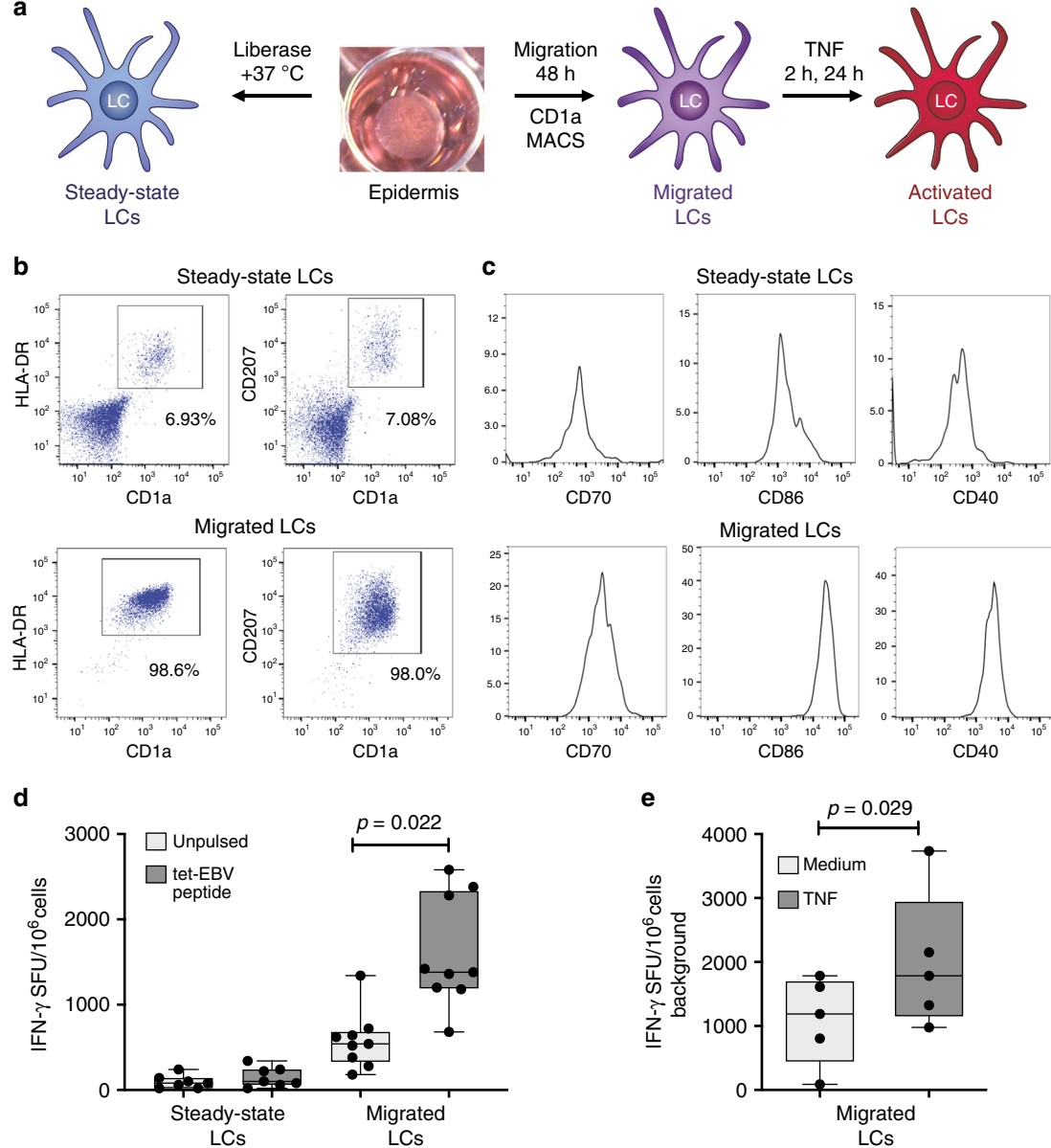

**Fig. 1 System for analysing human LCs and control of antigen cross-presentation. a** Schematic depicting isolation of primary human LCs. Split healthy skin was treated with dispase for 20 h to dissociate epidermis. Steady-state LCs were isolated from the epidermis by digestion with liberase TM or migrated from the epidermal sheets for 48 h in cell culture medium and stimulated with TNF (24 h) to induce their activation. **b** Flow cytometry assessment of steady-state and migrated LC. LCs were enumerated as CD207/CD1a/HLA-DR high cells. A representative example of $n > 5$ independent donors. **c** Flow cytometry assessment of activation markers expressed by steady-state and migrated LC. Co-stimulatory molecules critical for T-cell activation during antigen presentation (CD70, CD86 and CD40) were analysed in CD207/CD1a/HLA-DR high cells. A representative example of $n > 5$ independent donors. **d** IFN-γ secretion by an EBV-specific CD8 T-cell line stimulated with antigen presenting LCs in the context of MHC-I HLA-A2. Steady-state or migrated LCs were pulsed with 30-amino acid peptides containing EBV epitope (dark grey). Pulsed or unpulsed (light grey) LCs were stimulated with TNF and then assayed for IFN-γ secretion. ELISpot assay, $n = 5$ independent experiments in triplicate, paired $t$ test, box and whiskers show min and max value, line at median. Source data are provided as a Source Data file. **e** IFN-γ secretion by EBV-specific CD8 T-cell line stimulated by migrated LCs pulsed as in panel **d**. IFN-γ secretion was measured with (black) or without (grey) TNF stimulation. ELISpot assay, $n = 5$ independent experiments in triplicate, paired $t$ test, box and whiskers show min and max value, line at median. Source data are provided as a Source Data file.

a 30-amino acid peptide, comprising a 9-amino acid HLA-A2-restricted GLC epitope from Epstein Barr Virus protein BMLF and stimulated with TNF. We have previously demonstrated that the cross-presentation of the GLC epitope to antigen-specific CD8 T cells was critically dependent on the ability of LCs to take up and actively process the 30AA peptide for presentation[2]. LC migration from the skin induced their ability to cross-present antigens as measured by IFN-γ release from a GLC peptide-

specific HLA-A2 CD8 T-cell line (Fig. 1d; Supplementary Fig. 1b). TNF signalling further enhanced the ability of migrated LCs to cross-present the same antigen (Fig. 1e). We note that in the presence or absence of TNF, cells fixed with glutaraldehyde prior to antigen pulsing did not activate cognate CD8 T cells. In contrast, fixation did not affect presentation of a short peptide, which could be externally loaded onto the MHC molecules. Furthermore, fixing LC post pulsing but before co-culture with T

lymphocytes, reduced their ability to activate CD8 T cells[2]. This is consistent with the inhibition of intracellular protein trafficking and antigen processing by glutaraldehyde fixation. Thus, LC migration upregulates CD4 and CD8 T-cell co-stimulatory molecules and their antigen presentation as well as cross-presentation capabilities, the latter is augmented by TNF signalling.

**Genomic programming of human LCs for antigen presentation.** To gain insights into the genomic programming of migrated LCs, we analysed their transcriptome using bulk RNA-sequencing. The antigen processing and presentation genes were quantitatively amongst the highest expressed genes in migrated LCs, and are therefore designated as the core LC transcriptional programme (Fig. 2a). This confirmed and extended our previous analysis using DNA microarrays[22]. We next compared the expression of genes in migrated LCs with previously reported signatures of DCs, including those in the Reactome database and reported by Artyomov et al.[29]. These were compiled into antigen processing and antigen cross-presenting molecular signatures (Fig. 2b; Supplementary data 1, Supplementary Fig. 2a). In agreement with previously published data[29], the gene signature encoding antigen processing and presentation in different populations of human skin and blood-derived DCs was recapitulated in human LCs, suggesting the existence of a shared transcriptional programme (Fig. 2b; Supplementary Fig. 2a, b). While 53 genes shared between all three subsets encode for proteasome structure (41 genes, FDR $P = 7.32^{-93}$), protein catabolic process, (FDR $P = 8.164^{-100}$) and antigen presentation to class I (FDR $P = 1.324^{-95}$), 287 genes shared between LCs and other cross-presenting DCs were involved in metabolic processes (FDR $P = 5.39^{-19}$), including NADH dehydrogenase activity (FDR $P = 2.723^{-11}$). Notably, the core LC genomic programme was accompanied by high levels of expression of genes encoding ubiquitin protease activity (Fig. 2a). Accordingly, 64/66 genes shared between LCs and the antigen processing signature encoded protein ubiquitination components (FDR $P = 1.340^{-83}$). Importantly, migration from the epidermis also induced high levels of SQSTM1 and TRIM21 proteins, key components involved in antigen processing (Fig. 2c).

To uncover genes whose regulated expression could enhance the ability of migrated LCs for antigen cross-presentation, we performed RNA-seq by stimulating with TNF for 2 or 24 h. As shown before, TNF signalling leads to enhancement of LC antigen cross-presentation (Fig. 1e). Although, the overall transcriptional programme remained relatively stable under these activation conditions, 1156 genes were significantly differentially regulated by TNF (EdgeR, FDR < 0.05, |LogFC| > 1). Transcript-to-transcript clustering (BioLayout Express3D, $r = 0.80$; MCL = 1.7) identified seven kinetic clusters with $n > 8$ genes; three main clusters were characterised by gene expression peaks at 0, 2 and 24 h (Supplementary Fig. 2c–e, Supplementary Data 2). Gene ontology analysis indicated a consistent shift of the transcriptome towards a more activated LC phenotype; this included reduction of adhesion, enzymatic and trans-membrane signalling with the enhancement of immune functions (Supplementary Fig. 2c, Supplementary Data 2). Two waves of gene activation could be distinguished: an early wave, including the CD40 and CD83 genes, involved in T-cell interactions and a late wave, including *PSME2* and *TRIM21, 22* involved in antigen processing and protein ubiquitination (Fig. 2d, e; Supplementary Data 2). In agreement with our microarray analysis[22], the late wave included components of immunoproteasome (*PSME2*, *ERAP2)* genes involved in intracellular antigen trafficking between the cell membrane, the endosomal compartment and autophagosome

(*CAV1*, *SQSTM1*) (Fig. 2d, e). Interestingly, this programme also included many members of the superfamily of tripartite motif-containing (TRIM). TRIM proteins are E3 ubiquitin ligases involved in membrane trafficking, protein transport and protein degradation in proteasomes, crucial for many aspects of resistance to pathogens, and reported in the protection against lentiviruses[48,49]. High levels of SQSTM1 and TRIM21 were sustained during stimulation with TNF, together with high levels of expression of co-stimulatory molecules (Supplementary Fig. 2g, h)[2]. Thus, as a consequence of TNF stimulation, the LC transcriptome was highly enriched in genes involved in antigen processing and MHC-I-dependent antigen presentation (Supplementary Fig. 2c–g).

**Coupling of metabolism and antigen presentation in LCs.** Next, we used scRNA-seq to analyse the transcriptomes of migrated LCs (>90% of CD1a + , HLA-DR + cells). ScanPy analysis of 950 cells clearly identified three major cell clusters (#1–3), confirming LC identity of 916 cells (96.7%) in (ARCHS4 tissues, FDR $P = 4.446 \times 10^{-30}$, Fig. 3a, Supplementary Fig. 3a–d). The minor clusters 4 and 5 comprise cells expressing markers of melanocytes and T cells (Supplementary Fig. 3a, c).

Interestingly, the key genes defining common LC features were *HLADR* and *CD74* that encoding antigen presenting components, together with *TMSB4X*, which is involved in actin polymerisation, cell motility and cytoskeleton re-organisation and *MT-CO2, MT-CYB* that encode mitochondrial enzymes (Supplementary Fig. 3c, d, Supplementary Data 3). Analysis of the pseudotrajectories of cells in clusters 1–3 indicated progression of maturation (Cluster 3 → Cluster 1 → Cluster 2) with an enrichment for expression of genes involved in antigen processing and presentation (*PSMD4*, *PSMD7*, *UBE2L3* cluster 2, Fig. 3b–d; Supplementary Fig. 3d). Cells in clusters 1 and 2 also displayed expression of a higher proportion of genes involved in oxidative phosphorylation (Fig. 3c, d). The enrichment in genes encoding components of oxidative phosphorylation including cytochrome oxidase function is likely to be highly important for LC biology. The coupling of oxidative phosphorylation with efficient priming of immune responses has been reported previously for murine DCs, increased catabolism was shown to be essential for DC tolerogenic function[50]. Furthermore, mitochondrial membrane potential and ATP production enhances the phagocytic capacity of murine CD8 DCs, augmenting their cross-presentation at a late stage[51]. Thus, migrated LCs appear to be optimally transcriptionally programmed for priming of tolerogenic CD4 and protective CD8 T-cell responses.

We next compared the single-cell LC transcriptomes with the recently described sub-populations of human blood monocytes and DCs (GSE94820[52]), using CellHarmony[53] and SCmap[54] tools. Both mapping strategies confirmed that the majority of LC single-cell transcriptomes were distinct from the conventional DC1, and in contrast resemble cDC2 and cDC3 (Supplementary Fig. 3e).

**Enrichment of IRF composite elements in activate LC promoters.** To identify transcription factors involved in the genomic programming of human LCs, we perform chromatin profiling and transcription factor motif enrichment analyses. This approach enables the inference of transcriptional regulatory sequences and the transcription factors that are acting to control gene activity in distinct cell types and cell states[40,55]. Notably, tri-methylation of lysine 4 of histone 3 (H3K4Me3) marks active promoters[56], whereas acetylation of lysine 27 of histone 3 (H3K27Ac) marks active transcriptional enhancers; the latter have been postulated to be the primary determinants of

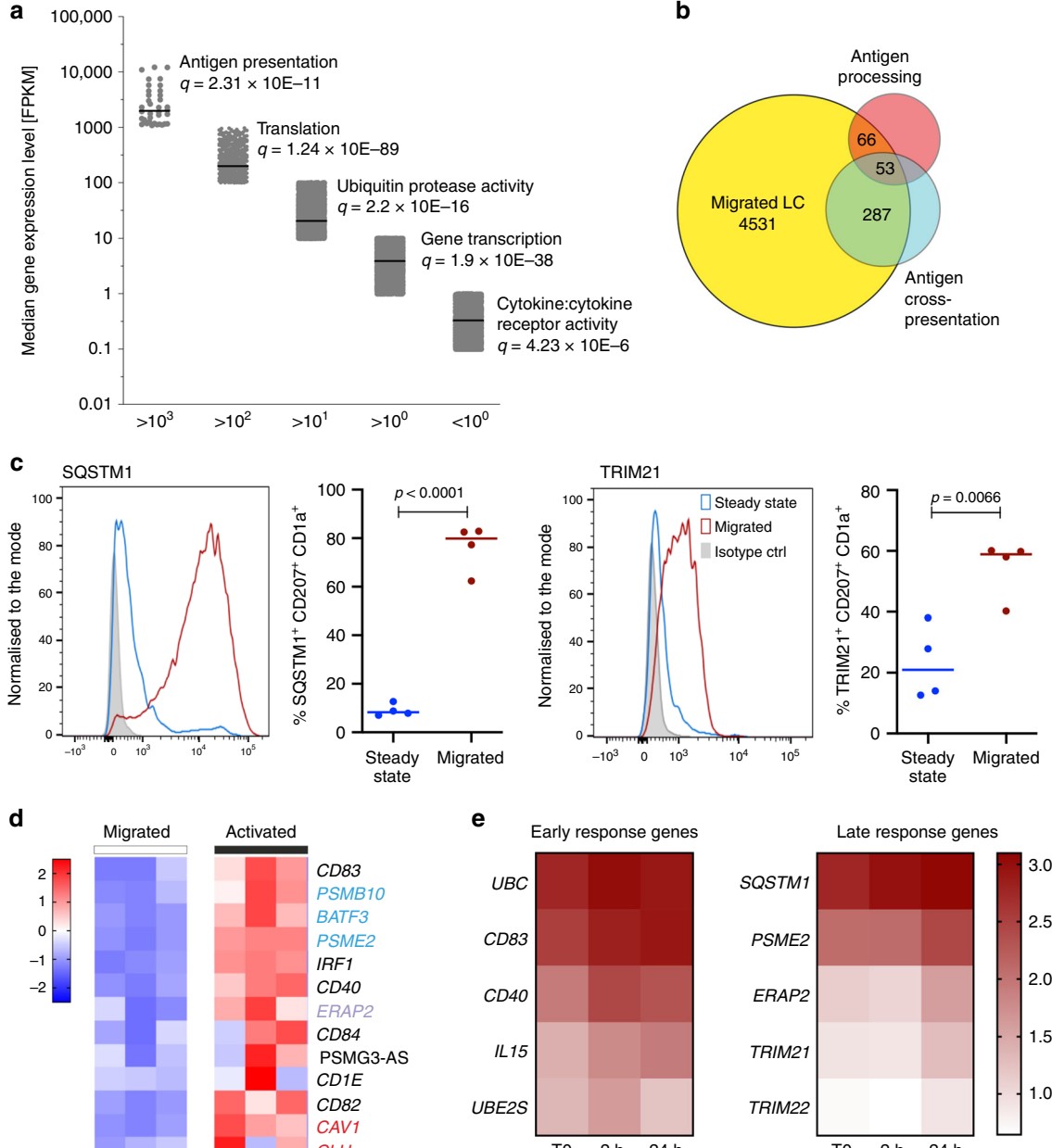

**Fig. 2 Transcriptional programming of migrated human LCs. a** Dominant biological processes and pathways enriched in genes expressed at varying levels in steady-state migrated LCs. Gene ontology analysis for each expression level (Fragments Per Kilobase of transcript per Million mapped reads, FPKM) interval determined by RNA-seq was performed using ToppGene on-line tool[78]. Line denotes median value in the interval. Top unique Biological processes are shown for each interval, significance denoted by FDR (Benjamini-Hochberg) corrected P-value is shown. The x-axis shows consecutive cut-offs for each interval in gene expression levels. Source data are provided as a Source Data file. **b** Overlaps between reported cross-presentation (373 genes) and antigen processing (212 genes) signatures, and genes expressed in migrated LC > 10 FPKM. **c** Intracellular expression of SQSTM1 and TRIM21 measured by flow cytometry. Steady-state (blue) and migrated (red) LCs. Representative histograms followed by quantitative analysis $n = 4$ independent donors, unpaired $t$ test, line denotes median value. Source data are provided as a Source Data file. **d** TNF stimulation (24 h) of human LCs induces genes involved in antigen trafficking (red), processing (purple) and cross-presentation (blue). Expression levels of three biological replicates (TMM normalised gene expression levels, scaled in rows). Source data are provided as a Source Data file. **e** Enrichment of immune activation genes upregulated during a time course of TNF stimulation: left: early induced genes, peak expression at 2 h, Cluster 3, right: late induced genes, peak expression at 24 h. Median of three biological replicates, normalised expression levels. Stars denote genes belonging to antigen presentation in class I, MSigBD, Broad Institute. Source data are provided as a Source Data file.

tissue-specific gene expression[57,58]. Thus, we performed ChIP-seq to analyse the genome-wide landscape of H3K4Me3 and H3K27Ac in migrated and TNF-activated LCs. Three independent samples of human migrated LCs exhibited a highly conserved histone modification landscape across the genome with overlapping H3K4Me3 peaks in 95% of the marked regions

(Supplementary Data 4). Analysis of H3K27 acetylation peaks in the same LC preparations shows 78% to be shared across all three samples (Supplementary Data 5). Of 13,402 H3K4Me3 peaks that were common to migrated LCs (associated with 11,665 unique genes), over 92% were mapped to promoter regions (Supplementary Data 4). In contrast, while 62% of H3K27Ac peaks were

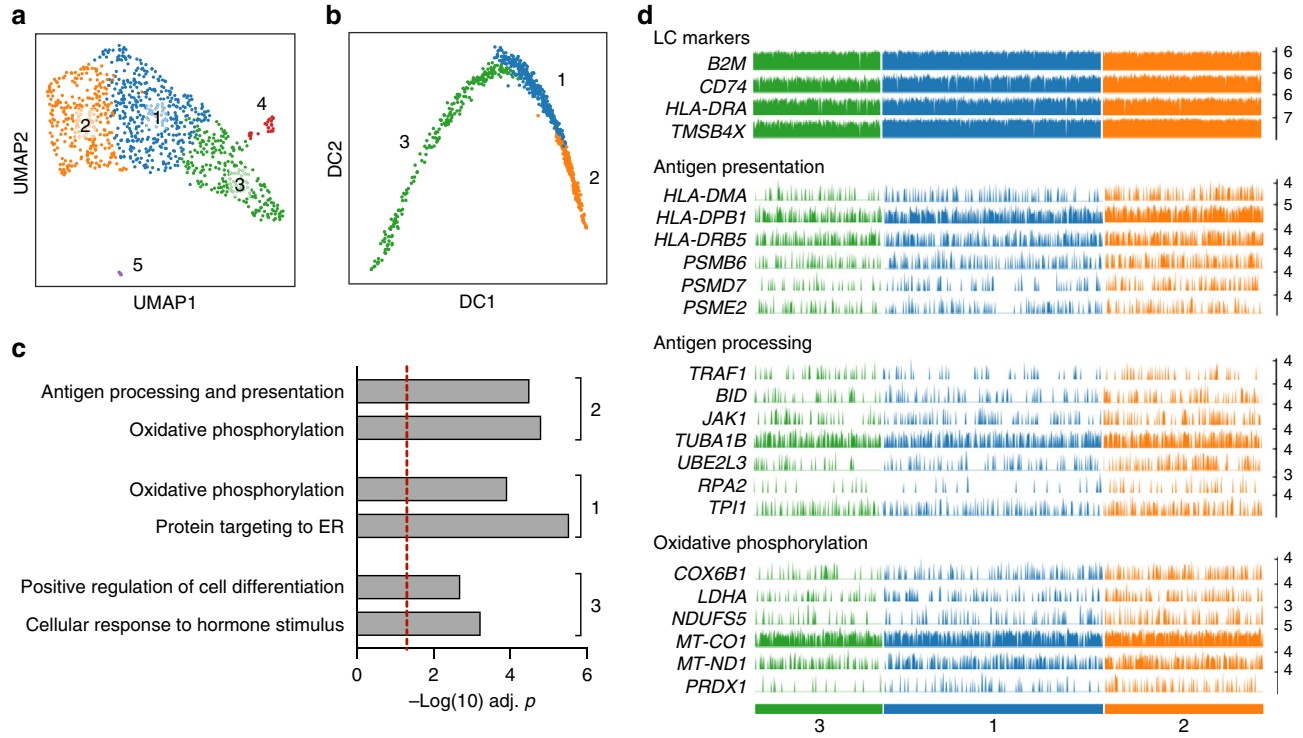

**Fig. 3 scRNA-seq analysis of migrated human LCs. a** UMAP plot of 950 migrated LCs (ScanPy, Leiden $r = 0.2$, n_pcs $= 4$, n_neighbours $= 10$, 2464 highly variable genes (min_mean $= 0.0125$, max_mean $= 6$, min_disp $= 0.6$) defines three major clusters of LCs. **b** Pseudotrajectory analysis of the transcriptomes of 950 migrated LCs (ScanPy, diffmap: Leiden $r = 0.2$, n_pcs $= 4$, n_neighbours $= 10$). Cells are colour coded for clusters as in panel **a**. **c** Gene ontology analysis for marker genes ($n = 100$) representative of indicated cluster, performed using ToppGene. –log(10) Benjamini-Hochberg corrected $P$-values are shown for cluster-specific processes. Source data are provided as a Source Data file. **d** Barplots displaying frequency and amplitude expression of indicated gene transcripts. Bars are colour coded for cells as in panel **a**. Representative uniformly expressed genes characteristic of LCs (top panel), genes involved in antigen presentation and processing (middle panels) and genes functioning in oxidative phosphorylation (bottom panel) are displayed. Each bar shows CPTT (counts per ten thousand) normalised expression level of indicated transcript in a given LC.

positioned either within the promoter region or upstream, a significant proportion of these peaks were distributed within inter- or intragenic regions (Supplementary Data 5). This distribution was expected for intergenic or intronic enhancers that function at large distances from the promoters on which they act.

Focusing the analysis on genes associated with immune function (InnateDB: Immunome collection[59], we identified 290 immune genes with active (H3K4Me3) promoters. These were highly enriched in genes encoding receptor binding and activation (in particular able to respond to TNF cytokine family signalling (FDR $P = 1.2 \times 10^{-12}$), and genes involved in antigen processing and presentation (FDR $P = 10^{-22}$). As noted above, considerable overlap between H3K4Me3 and H3K27Ac marks was apparent at promoter regions. This indicated that migrated LCs are pre-programmed for efficient antigen processing and presentation (Supplementary Fig. 4a–c). In concordance with the observed gene expression pattern, histone marks were very low or absent on genes involved in innate inflammatory responses, such as production of cytokines (Supplementary Data 4, 5). To analyse whether genomic programming of human migrated LCs was similar to other known cell types including monocytes, DCs and macrophages, we compared our ChiP-seq profiles to a large collection of publically available genomics datasets (see the Methods section). Surprisingly, the chromatin landscape of migrated LCs was strikingly similar to that of macrophages and CD14+ monocytes, and significantly less strongly correlated with that of dendritic cells (Supplementary Data Table 6).

Stimulation with TNF preserved H3K27Ac acetylation in genes underpinning LC activation (Fig. 4a, b). Two hours after TNF

exposure, the high levels of H3K27 acetylation, observed in steady-state LCs (Fig. 4a) were maintained for 90% and 92% of genes induced during the early (cluster 3) and late (cluster 2) waves of responses, respectively (Fig. 4a). Moreover, TNF signalling enhanced H3K27Ac levels in over 50% of genes (Supplementary Fig. 4a). In line with the transcriptome changes, genes with enhanced H3K27Ac marks at 2 h encoded innate immune processes including leukocyte activation (FDR $P$-value $= 2 \times 10^{-19}$) and co-stimulation (FDR $P$-value $= 7.99 \times 10^{-16}$). A significant proportion of genes (286) within this programme was associated with ubiquitin-mediated proteolysis, highlighting the importance of this process for LC function (FDR $P = 3.8 \times 10^{-85}$). In contrast, genes involved in cell cycle and motility were characterised by reduced histone acetylation marks. It is worth noting that the majority of histone marks induced by TNF signalling were readily detected prior to activation (Fig. 4a, b; Supplementary Fig. 4a–c), suggesting that the migrated LCs were pre-programmed for rapid immune activation from a genomic standpoint.

To infer transcription factors directly controlling the transcriptional programmes in human LCs, we performed TF motif enrichment analyses, revealing an extremely high enrichment of the composite interferon regulatory factor-binding (IRF-binding) sequences[36] at the promoters of genes carrying H3K4Me3 and H3K27Ac marks (Fig. 4c). The ETS-interferon composite element (EICE) was the most frequent and significantly enriched TF motif in steady-state migrated LCs (47%, –log $P = 16,020$), while the AP-1-interferon composite element (AICE) and the interferon-stimulated response element (ISRE/IRF1) were present at

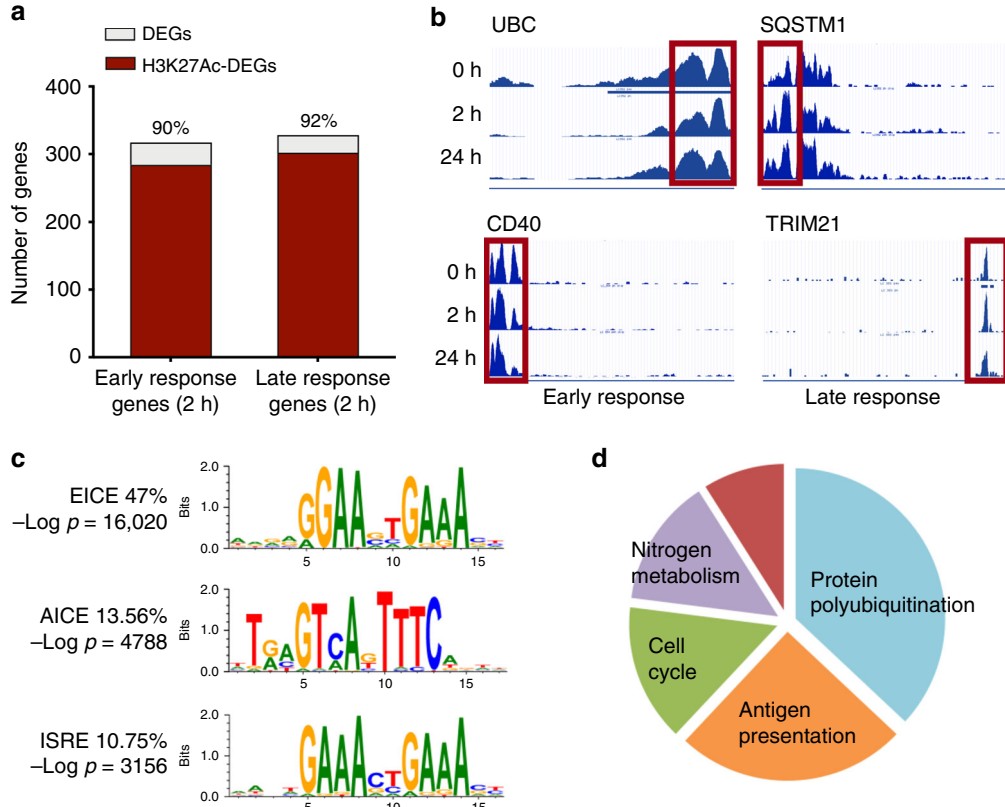

**Fig. 4 Chromatin landscape of migrated LCs enriches for EICE, AICE and ISRE motifs.** Human steady-state migrated LCs were subjected to whole-genome chromatin profiling of H3K4me3 and H3K27Ac. **a** Proportion of DEG with H3K27Ac mark at 2 h in clusters of co-expressed genes upregulated early (2 h, clusters 3) and late (24 h, cluster 2) following stimulation with TNF. Changes in H3K27Ac acetylation were calculated using MANorm algorithm embedded in BioWardrobe tool. Genes were filtered to include unique common entry across the biological replicates (consensus value from $n = 3$ independent donors). Genes with detected changes in acetylation were intersected with DEGs identified by EdgeR analysis. Source data are provided as a Source Data file. **b** UCSC genome browser tracks of H3K27Ac mark changes in human migrated LCs over the time course of stimulation with TNF. Early ubiquitin C, UBC (top) and CD40 (bottom) and late sequestosome SQSTM1 (top) and TRIM21 (bottom) induced genes. Red rectangle denotes promoter site, a representative example. **c** Promoter sites of genes acetylated at H3K27 in migrated human LCs are enriched in IRF-binding composite DNA elements. EICE is the top human motif enriched at steady-state LCs (HOMER de novo motif detection analysis, median –log P-value shown). **d** Peaks H3K4Me3 and H3K27Ac T0 datasets were scanned for ISRE/AICE/EICE binding motifs. In total, 1193 consensus transcripts (present in all three biological replicates with both chromatin marks) were identified. Biological processes enriched in those genes were detected using ToppGene based on FDR-corrected P-values for GO categories and collapsed to overarching categories. Source data are provided as a Source Data file.

frequencies of 13.6% ($-\log P = 4788$) and 10.3% ($-\log P = 3.156$), respectively (Fig. 4c; Supplementary Data 7). Analysis of biological processes enriched for genes carrying H3K4Me3 and H3K27Ac marks in their promoter regions and predicted IRF4, 8 binding motifs demonstrated that over 60% were involved in protein polyubiquitination, or antigen processing and presentation (Fig. 4d; Supplementary Fig. 4d). Thus, transcription and chromatin profiling coupled with TF motif analysis strongly suggested that IRF4 or IRF8 in conjunction with PU.1 and BATF3, binding to EICE and AICE motifs, respectively, could programme the expression of a large set of genes in human LCs.

**LC migration and maturation associate with IRF4, not IRF8.**
Given critical roles for specific members of the IRF, ETS and AP-1 family transcription factors in antigen presentation in conventional DCs[42,60], and the enrichment of EICE and AICE composite motifs in active chromatin regions of LCs, we analysed the expression of IRF4, IRF8, PU.1, SPIB, cJUN, JUND and BATF3, in steady-state and migrated LCs. Whereas IRF4 and BATF3 proteins were expressed in steady-state LCs surprisingly there was no detectable expression of IRF8 (Fig. 5a; Supplementary Fig. 5a–f). Importantly, migration out of the skin further

induced IRF4 expression (Fig. 5a, b). In contrast, IRF8 protein remained undetectable in migrated LCs (Fig. 5a). Analysis of transcripts for these transcription factors was in keeping with the expression of their proteins (Fig. 5c; Supplementary Fig. 5a–f). We note that transcripts encoding *IRF4*, *PU.1* (Spi1) and *cJUN* were most highly expressed in LCs, and were at least an order of magnitude greater than those encoding *IRF8* and the PU.1 paralog *SpiB*. TNF stimulation maintained the expression of BATF3 protein while downregulating IRF4. Notably, IRF8 protein remained undetectable over the time course of TNF stimulation, in spite of low level upregulation of IRF8 mRNA (Fig. 5c, d; Supplementary Fig. 5f). Thus, the genomic programming of human LCs and also their ability to cross-present antigens appears to be IRF8 independent. Instead, the results strongly suggest that human LCs depend on IRF4 along with PU.1 and BATF3 for transcriptional programmes underpinning their contextual functions, including expression of antigen cross-presentation genes.

**IRF4 balances LC maturation and inflammatory signalling.**
Given that IRF4 is upregulated when LCs migrate from the epidermis (Fig. 5a, b), a process associated with their maturation as

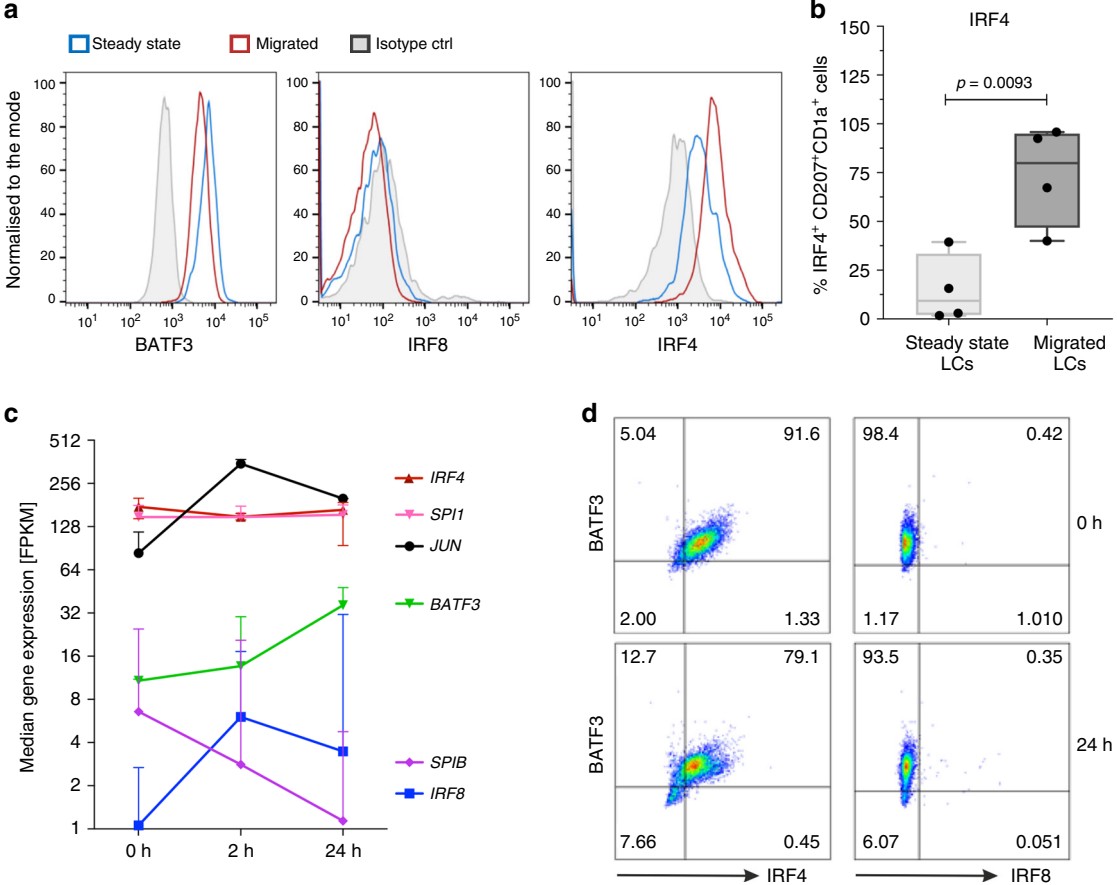

**Fig. 5 Human LCs upregulate expression of IRF4 upon migration, but lack IRF8. a** IRF4, but not IRF8, protein expression is upregulated in BATF3-positive LC during migration from the epidermis. A representative FACS analysis of 3–5 independent donors, gates set using isotype controls for each antibody (nuclear staining for IRF4, IRF8 and BATF3). **b** IRF4 protein expression in steady-state vs migrated LCs. IRF4+ LCs (%) as measured by flow cytometry, median ± range, $n = 4$, three paired samples from independent donors, unpaired $t$ test, box and whiskers show min and max value, line at median. Source data are provided as a Source Data file. **c** Transcript levels of key transcription factors in migrated LCs before and after TNF stimulation (2 h, 24 h). FPKM values, median ± range of three biological donors are shown. Source data are provided as a Source Data file. **d** Expression of IRF4 and BATF3 in response to TNF signalling (24 h). Representative graphs of five independent donors, gates set using isotype controls for each antibody (Nuclear staining for IRF4, IRF8 and BATF3).

well as enhanced antigen cross-presentation (Fig. 1d), we wished to determine how perturbation of IRF4 impacts the genomic programming of migrated LCs. CRISPR-Cas9 editing with an IRF4 guide–Cas9 complex was used to knockdown IRF4 expression in migrated LCs (Fig. 6a, b; Supplementary Fig. 6a). Importantly, transfection of LCs by nucleofection did not lead to enhanced cell death (Supplementary Fig. 6b). The effect of genome editing was sustained at 72 h and 96 h post nucleofaction, however, the cell viability decreased with time both in the knockdown and in the control cells. scRNA-seq of 1000 control (wild-type, WT) and 1000 edited (knockdown, KD) cells confirmed significantly lower levels of IRF4 expression at the mRNA level, and separation of transcriptomes of KD and control cells in the uMAP plot (Fig. 6c; Supplementary Fig. 6c).

Comparison of transcriptomes of WT and KD LCs demonstrated importance of IRF4 for regulation of key processes in LCs. IRF4 KD LCs were impaired for expression of genes involved in myeloid leukocyte activation, including *LYZ*: antimicrobial function, *CTSH*: antigen processing and *WFDC21P*: a long non-coding RNA implicated in DC differentiation, FDR $P = 1.25^{-4}$) as well as cellular metal ion homoeostasis (FDR $P = 1.22^{-4}$). Interestingly, expression of genes encoding ubiquitin pathway components (ubiquitin protein ligase binding, FDR $P = 2.80^{-2}$) was also diminished. These genes showed strong overlap with

genes carrying H3K4Me3 and H3K27Ac marks (Supplementary Data 4, 5, 8). Strikingly, IRF KD LCs manifested increased expression of genes encoding responsiveness to cytokines (FDR $P = 1.52^{-13}$) and cellular stress, including oxidative stress (FDR $P = 8.98^{-12}$), with induction of *NFkB1, MAP4K4, NFL2F1* and *NFL2F2* (Fig. 6d–f, Supplementary Fig. 6c, d, Supplementary Data 8, 9). Thus IRF4 positively regulates genes involved in LC maturation while attenuating those involved in inflammatory cytokine and oxidative stress signalling.

## Discussion

Since their discovery by Paul Langerhans in 1864, LCs have been puzzling scientists. Despite being the longest studied antigen presenting cell population, and being considered the stereotypical dendritic cells, their place in the innate immune cell spectrum has remained elusive. This is due to the difficulties in studying LC, arising from paucity of in vitro models, controversy around defining a LC progenitor, and the differences between human and murine skin immunology[61]. To address these challenges, we analysed primary human LCs, using bulk and single-cell RNA-seq as well as H3K4Me3 and H4K27Ac ChIP-seq and couple these genomic and epigenomic profiles with LC phenotypic and functional characteristics. Our extensive transcriptomic and

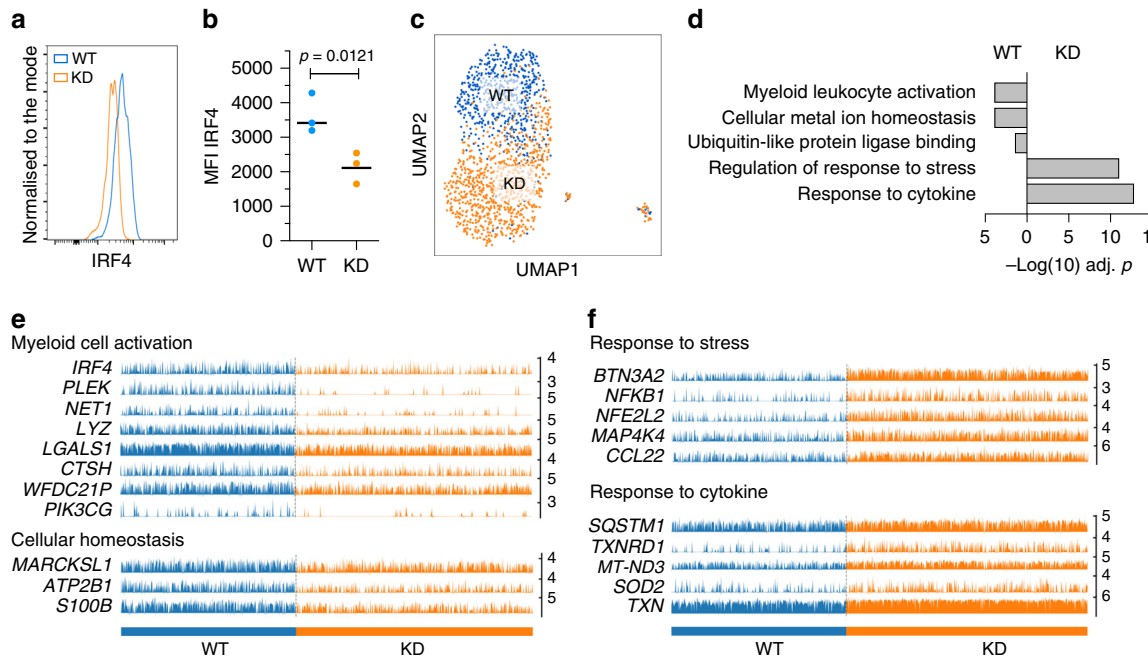

**Fig. 6 IRF4-mediated transcriptional programming of human LCs.** Knockdown (KD) of IRF4 using CRISPR-Cas9 editing. An IRF4 guide–CAS9 complexes were introduced by nucleofection into migrated LCs. Indicated analyses were performed 48 h after nucleofection. **a** Intranuclear expression levels of IRF in CRISPR-Cas9 edited (orange) and control (blue) migrated LCs measured using flow cytometry in CD207+ CD1a+ live LCs. A representative example of $n = 3$ independent donors. **b** Quantification of the intranuclear expression levels of IRF4 in CRISPR-Cas9 edited (orange) and control (blue) migrated LCs. Mean fluorescence intensity (MFI) of IRF4-expressing CD207+ CD1a+ live LCs shown, $n = 3$ independent donors, paired $t$ test, box and whiskers show min and max value, line at median. Source data are provided as a Source Data file. **c** scRNA-seq analysis of WT and IRF4 KD migrated LCs. UMAP plot of 1484 LCs (WT:617 cells, blue, KD, 867 cells, orange) (filtering setting gene count 500–10,000 per cell, expression of mitochondrial genes <0.2, Leiden $r = 0.2$). **d** GO processes and pathways differentially enriched in control LC (WT, left) versus IRF4 CRISPR-Cas9 edited LCs (IRF4 KD, right). DEG analysis was performed using Single TK package, scDiffExlimma algorithm, scnorm data, FDR-corrected $P$-value used as a significance measure. Gene ontology enrichment: ToppGene tool, Biological Processes and Molecular Function. Source data are provided as a Source Data file. **e** Barplots of genes expressed at higher levels in WT LCs. Each bar shows CPTT normalised expression level of indicated gene in a given WT LC cell (blue) or IRF4 CRISPR-Cas9 edited LC (orange). IRF4 plus top five genes by FDR-corrected $P$-value and five genes representative for biological processes enriched in WT are displayed. **f** Barplots of genes expressed at higher levels in IRF4 KD LCs. Each bar shows CPTT normalised expression level of indicated gene in a given WT LC cell (blue) or IRF4 CRISPR-Cas9 edited LC (orange). Top five genes by FDR-corrected $P$-value and five genes representative for biological processes enriched in KO are displayed.

epigenomic analyses of primary human skin-derived LCs reveals three striking features: (i) the pre-programming of LC chromatin and transcriptome landscapes, (ii) the importance of migration from the epidermis for activation of the LC maturation and (iii) the key role of a major immune transcriptional regulator IRF4 rather than its paralog IRF8. We suggest all three features are likely to be inter-connected, and dictated by the localisation of LCs in the epidermis and their functions in maintaining immune homoeostasis[62].

By profiling the H3K4Me3 and H3K27Ac histone modifications across the LC genome, we were able to analyse the chromatin landscape of human migrated LCs for the first time, and document that their genomic programme encompassing antigen-presentation genes is poised for efficient expression during, or before their migration from the epidermis. TNF, a pro-inflammatory cytokine produced in the skin plays a critical role in inducing LC immunogenic function and ability to present antigens. Stimulation with TNF, significantly enhanced the pre-existing transcriptional programme, further confirming that the LCs are fully committed for efficient antigen presentation as well as cross-presentation. Such genomic programming, realised at the level of transcriptional enhancers, could be both developmentally and environmentally specified[63].

We have delineated three distinctive IRF-binding motifs, EICE, AICE and ISRE as key regulatory elements associated with

expression of the LC transcriptional programme. Classically, EICE and AICE have been shown to be bound by IRF4 or IRF8, in combination with their transcriptional partners from either ETS or AP-1 transcription factor families[36,64–67]. Our analysis of IRF4 and IRF8 protein expression and their transcriptional binding partners clearly demonstrates that IRF8 protein is not detectably expressed in human LCs and is thus dispensable for their function, including cross-presentation of exogenous anti-gens to CD8 T cells. IRF8 has been implicated as a key tran-scription factor in murine CD8alpha + DCs regulating genes involved in cross-presentation through interaction with BATF on composite elements (AICE) in the promoters of target genes[42,45]. By contrast, recent reports in other cell types, such as MoDCs[68], indicate that the same transcriptional programme can be regu-lated by IRF4[68]. Hence the high levels of IRF4 expressed in human migrated LCs are likely to be involved in the orchestration of this programme[68].

The analysis of LCs edited for expression of IRF4 using a CRISP-Cas9 system allowed us to directly test the importance of IRF4 in the transcriptional programming of human LCs. In concordance with studies by Chopin et al.[30,47], IRF4 was not critical for LC survival. However, we demonstrated that LC genomic programming was critically dependent on IRF4 func-tion. Knockdown of IRF4 resulted in the impaired expression of genes involved in LC activation, homoeostasis as well as

ubiquitin-dependent antigen processing pathways. Strikingly, knockdown of IRF4 resulted in the increased expression of genes encoding multiple components of the NFκB and NF2EL2 pathways that control responsive to inflammatory cytokines, such as TNF and oxidative stress. Thus, IRF4 appears to dampen the response of LCs to inflammatory cytokines and in so doing may promote tolerogenic responses in the skin. Such a function for IRF4 in dendritic cells in priming Treg responses has been previously observed in murine system[40]. The balance between efficient antigen presentation and responsiveness to inflammatory signalling appears to be critical for LC biology. One of the key functions of LCs is maintenance of peripheral tolerance in situ[5] and through continuous trafficking of cutaneous self-antigens to regional lymph nodes[26,27]. In contrast, upon exposure to pro-inflammatory cytokines, cross-presentation of antigens by LCs plays important role for adaptive immune responses[10,22,24]. We note that, in parallel with enhancing LC ability to stimulate CD8 T cells, TNF signalling in LCs downregulated the expression of IRF4 protein. Thus, we propose that a reciprocal feedback inhibition loop between inflammatory cytokines and IRF4 may be critical for balancing tolerogenic versus immunogenic LC responses in the epidermis[2,22,69–72].

The notable lack of IRF8 expression in LCs, consistent with inactive chromatin at the IRF8 locus, distinguishes LCs from both macrophages and DCs and likely contributes to their discrete role in epidermal homoeostasis. Independence of LCs from IRF8 could represent a mechanism for their plasticity, enabling them to be adapted to their environmental niche. IRF8 has been shown to regulate production of pro-inflammatory cytokines in DCs[73,74], and macrophages, the latter contributing to chronic inflammation[75]. We propose that by utilising IRF4 rather than IRF8, LCs uncouple the cross-presentation of antigens from production of pro-inflammatory mediators, and thus prevent excessive inflammatory responses and promote epidermal homoeostasis. It is plausible that the increase in IRF4 expression during LC migration out of the epidermis makes LCs immunocompetent, in a highly controlled manner, activating IRF4-coordinated transcriptomic programmes centred around efficient antigen presentation, but independent of NFkB1/MAPK4K signalling. These observations along with our earlier analysis of IRF4 function in murine DCs[40] lead us to propose a context-dependent model of LC activation, where loss of contact with epidermal structural cells upregulates IRF4 expression and initiates an LC activation programme that promotes tolerogenic T-cell responses and immune homoeostasis. Signalling by pro-inflammatory cytokines attenuates IRF4 expression and elicits a transcriptomic programme that more effectively primes immunogenic T-cell responses.

## Methods

**Cell isolation and stimulation with TNF.** Skin specimens and blood samples were acquired from healthy individuals after obtaining informed written consent with approval by the Southampton and South West Hampshire Research Ethics Committee in adherence to Helsinki Guidelines [ethical approvals: 07/Q1704/59, NRES 07 Q1704 46]. Split skin was obtained using graft knife and subjected to dispase (2U/ml, Gibco, UK, 20 h, + 4 °C) digestion of epidermal sheets. Migrated LCs were harvested after 48 h culture of epidermal fragments in full culture media (RPMI, Gibco, UK, 5% FBS, Invitrogen, UK, 100 IU/ml penicillin and 100 mg/ml streptomycin, Sigma, UK). Low-density cells were enriched using density-gradient centrifugation (Optiprep 1:4.2, Axis Shield, Norway[2] and purified with CD1a+ magnetic beads according to the manufacturer's protocol (Milenyi Biotec, UK). Migrated LCs were processed for RNA-seq and ChIP-seq experiment or immediately cryopreserved in 90% FBS (Gibco, UK), 10% DMSO (Sigma, UK). For genomic and transcriptomic analyses of activated LCs, fresh migrated LCs from three donors were stimulated with TNF (25 ng/ml, Miltenyi Biotec, UK) for 2, and 24 h (RNA-seq: $3 \times 10^5$ cells/donor/time point, ChIP-seq: $1.5–2 \times 10^6$ cells/donor/time point, paired samples from the same donor for RNA-seq and ChIP-seq). Steady-state LCs were enzymatically digested from the epidermal sheets using Liberase[TM] TM research grade (Roche, UK, 2 h at 37 °C).

**Antigen cross-presentation assay.** Cells were pulsed with 10 μM proGLC (FNNFTVSFWLRVPKVSASHLEGLCTLVAML; Peptide Protein Research, Fareham, UK) for 24 h, supplemented with TNF (25 ng/ml, Miltenyi Biotec, UK) after initial 2 h. Human responder cells: PBMC from HLA-A2 individuals were isolated by Ficoll-Hypaque density-gradient centrifugation and co-cultured with 40 μM EBV peptide for 12 days in complete medium supplemented with 1% sodium pyruvate (Gibco, UK) plus 10% human serum (Sigma, UK). IL-2 (100 IU/ml, Peprotech, UK) was added every 3 days. This method expands the pool of GLC-specific CD8 T cells to 30% (assessed by tetramer assay and ELISpot assay Polak et al.[2]). IL-2 was removed from the culture for 24 h prior to testing in ELISpot. For ELISpot assays, TNF matured and washed EBV peptide pulsed LCs ($1 \times 10^3$ cells) were co-cultured with GLC peptide-specific T cells ($5 \times 10^4$ cells/per well) for 20 h as per the manufacturer's protocol (Mabtech, Sweden).

**Flow cytometry.** All antibodies were used at pre-titrated, optimal concentrations. For surface staining of live cells, buffer containing 5% FBS and 1% BSA was used for all antibody staining. FACS Aria flow cytometer (Becton Dickinson, USA) was used for analysis of human LCs for the expression of CD207, CD1a, HLA-DR (mouse monoclonal antibodies, CD1a, CD207:Miltenyi Biotech, UK and HLA-DR: BD Biosciences, UK). For transcription factor intranuclear staining, cells were permeabilised with Foxp3/Transcription Factor Staining Buffer Set (eBiosciences, UK) according to the manufacturer's protocol, and stained with monoclonal antibodies targeting IRF4, IRF8, BATF3, PU.1, cJUN, (IRF4:rat monoclonal, eBiosciences, UK, mouse monoclonal: IRF8, eBiosciences, BATF, R&D Systems, JUN Millimark, UK, PU.1 Biolegend, UK). IRF1 staining was done using rabbit polyclonal anti-human IRF1 antibody (Abcam, UK) following fixation with 80% methyl alcohol and permeabilisation with Tween20. Analysis was performed on live AQUA-negative (Invitrogen, UK), CD207+ /HLA-DR+ migrated or steady-state LCs, in comparison with appropriate isotype controls.

**RNA-seq.** RNA was isolated using RNeasy mini kit (Qiagen) as per the manufacturer's protocol. RNA concentration and integrity were determined with an Agilent Bioanalyser (Agilent Technologies, Santa Clara, CA). All the samples had a RNA integrity number of 7.0 or above, and were taken forward for labelling. RNA-seq libraries were generated from 300 ng of total RNA with an RNA Sample Prep Kit (Illumina) according to a standard protocol. The libraries were sequenced with Illumina HiSeq2500 in the DNA sequencing core of the Cincinnati Children's Hospital Medical Center. Each sample was used to generate $2 \times 10^7$ reads with 75-base pair paired-end sequencing.

**RT-qPCR.** The expression of chosen genes was validated with quantitative PCR, using the TaqMan gene expression assays for target genes: YWHAZ (HS03044281_g1), CAV1 (Hs00971716_m1), PSME2 (Hs01923165_u1), (Applied Biosystems, Life Technologies, Paisley, UK) in cells from independent donors. RNA extraction (RNeasy micro kit, Qiagen) and reverse transcription (High-Capacity cDNA Reverse Transcription Kit with RNAse inhibitors, Applied Biosystems, Life Technologies, Paisley, UK) were carried out accordingly to the manufacturer's protocol.

**ChIP-seq.** Purified migrated LCs were fixed with 1% formaldehyde for 15 min, and the reaction quenched with 0.125 M glycine. Chromatin was isolated by the addition of lysis buffer, followed by disruption with a Dounce homogeniser. Lysates were sonicated, and the DNA sheared to an average length of 100–200 bp (Covaris). Genomic DNA (Input) was prepared by treating aliquots of chromatin with RNase, proteinase K and heat to remove crosslinks, followed by ethanol precipitation. Pellets were resuspended, and the resulting DNA was quantified on a NanoDrop spectrophotometer. Extrapolation to the original chromatin volume allowed quantitation of the total chromatin yield. Genomic DNA regions of interest were isolated using 2.8 μg of antibody against H3K27Ac or H3K4Me3[76]. Complexes were washed, eluted from the beads with SDS buffer and subjected to RNase and proteinase K treatment. Crosslinks were reversed by incubation overnight at 65 °C, and ChIP DNA was purified by phenol–chloroform extraction and ethanol precipitation. Illumina sequencing libraries were prepared from the ChIP and input DNAs by the standard consecutive enzymatic steps of end-polishing, dA-addition and adaptor ligation using Truplex[TM]—FD prep kit (Rubicon Genomics, USA). After a final PCR amplification step, the resulting DNA libraries were quantified using Qubid, and 75 nucleotide single-end reads were sequenced Illumina HiSeq2500 in the DNA sequencing core of the Cincinnati Children's Hospital Medical Center.

**Drop-seq.** Highly purified human LCs (> 80% of CD1a+ HLA-DR+ ) were unbanked from cryo-storage, and processed on ice to enable the co-encapsulation of single cells with genetically encoded beads (Drop-seq[77]). Monodisperse droplets at 1 nl in size were generated using the microfluidic devices fabricated in the Centre for Hybrid Biodevices, University of Southampton. To achieve single-cell/single-bead encapsulation barcoded Bead SeqB (Chemgenes, USA), microfluidics parameters (pump flow speeds for cells and bead inlets, cell buoyancy) were adjusted to optimise cell-bead encapsulation and the generation of high quality cDNA libraries. Following encapsulation, ~4500

STAMPS (beads exposed to a single cell) from 1.2 ml of cell suspension were generated in the Faculty of Medicine University of Southampton Drop-seq Facility. Based on encapsulation frequencies and bead counts, up to 1000 STAMPS were taken further for library prep (High Sensitivity DNA Assay, Agilent Bioanalyser, 12 peaks with the average fragment size 500 bp). The resulting libraries were run on a shared NextSeq run ($1.5 \times 10^5$ reads for maximal coverage) at the Wessex Investigational Sciences Hub laboratory, University of Southampton, to obtain single-cell sequencing data.

**Bioinformatic analysis of sequencing data**. All sequencing data have been analysed using established bioinformatic pipelines. For full description, please see the Supplementary Methods.

**CRISPR-Cas9 gene editing**. Primary human migrated LCs were subjected to nucleofection (185,000 cells/20 μl P3 reagent per reaction, Lonza protocol EH104) with purified *S. pyogenes* Cas9 (spCas9) complexed with a modified single-guide RNAs (sgRNAs) targeting *IRF4* (Synthego, C*G*C*AGGCGCGUCUUCC AG*G*U*). sgRNAs had the following modifications to increase stability: 2′-O-methyl analogs and 3′ phosphorothioate internucleotide linkages at the first three 5′ and 3′ terminal RNA residues. Ribonucleoprotein complexes were prepared in 1:6 (vol:vol) ratio (protein to modified RNA oligonucleotide) in ddH₂O immediately prior to nucleofection. Following culture (24–96 h), cell viability and IRF4 expression at the protein level were assessed by flow cytometry. scRNA-sequencing using Drop-Seq encapsulation was carried out on 1000 control and 1000 edited primary LCs cells at 48 h time point.

**Reporting summary**. Further information on research design is available in the Nature Research Reporting Summary linked to this article.

## Data availability
Sequencing data for RNA-seq, scRNA-seq and ChIP-seq is stored in Gene Expression Omnibus database, submission number GSE120386.

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

## Acknowledgements

We are grateful to the subjects who participated in this study. We would like to thank Dr. Krista Dienger-Stambaugh, Cincinnati Children's Hospital Medical Center for technical help and Dr. Nathan Salomonis, for introduction to AltAnalyzer. We acknowledge the use of the IRIDIS High Performance Computing Facility and Flow Cytometry Core Facility, together with support services at the University of Southampton.

## Author contributions

M.E.P. and H.S.: intellectually conceived and wrote the paper; M.E.P., M.A.J., S.S., K.C. and Z.W.: biological experiments and flow cytometry, processing of RNA and chromatin; M.E.P., J.W., V.C. and C.W.: analysis and meta-analysis of bulk RNA-seq data; M.E.P., J.R., M.P., X.C. and M.W.: analysis of ChIP-seq data; M.A.J., P.S. and M.W.: reviewing of the paper; M.E.P., P.S., M.R.Z., J.W., A.V., J.D., B.M.A. and S.S.: processing cells for scRNA-seq, pre-processing and analysis of scRNA-seq data; S.S., L.N., G.W. and M.E.P.: CRISPR-Cas9 editing.

## Competing interests

The authors declare no competing interests.
