## [Peer Review File · Nature Communications]

Reviewers' comments:

Reviewer #1 (Skin APC functions)(Remarks to the Author):

Langerhans cells (LC) were originally defined as prototypic DC, and the processes of maturation and migration to draining LN for stimulation of naive CD4 and CD8 T cells considered a paradigm for DC behaviour in other tissues. However, our more recent understanding that LC are derived from macrophage lineage cells has led the field to question our understanding of LC function. Here the authors investigate the proteins that programme antigen presentation in LC under the control of the transcription factors IRF4 and 8 that are important for the development, and potentially function, of DC1 and DC2 respectively. The key finding is that in contrast to cDC1, IRF4, not IRF8, is important for the activation of genes associated with antigen processing and cross-presentation in LC and this process is initiated as LC migrate out of the epidermis.

This is a comprehensive study in which the authors have used a range of tools with which to directly investigate the factors controlling antigen presentation in human LC. I have some concerns however that should be addressed:

1. The study relies heavily on interpretation of transcriptomic and ChIPseq data to infer conclusions about the links between IRF4 and antigen cross-presentation. There is some mechanistic data using the IRF4^{-/-} mouse, but these experiments are very limited and shown in the supplemental data (see also my comment 2.). The paper would benefit from experiments such as those used by the group for their 2014 Nature Immunology paper (DOI: 10.1038/ni.2795) - e.g What is the impact of the conditional deletion of IRF4 in LC? Do these cells still present and cross-present antigen in vitro and topically applied antigen in vivo? What is the effect on formation of peptide-MHC I/II complexes in IRF4^{-/-} LC?

2. IFN γ production by recently primed OT-I cells is a poor readout for in vitro assays with DC /LC (Supp figure 4), and as a result the data are not robust. T cell division (e.g. CFSE dilution) would be a clearer read-out (again as used in the 2014 paper) with down-regulation of CD62L or up-regulation of CD25/CD69. It would be helpful to compare wild-type and IRF8-deficient DC1 in these assays.

I also found the paper difficult to read, and suggest that the authors consider re-writing it if it were to be published in a broad science paper such as Nature Communications. For example:

1. The authors frequently allude to a fundamental role for IRF8 in cross-presentation by DC and this assertion is key to their conclusion that LC cross-present antigen despite an absence of IRF8. However, I am unaware of work showing a direct role for IRF8 for the cross-presentation machinery in DC1. A number of references are cited in the introduction, but they all refer to the interaction between IRF4 and presentation on MHCII to CD4 T cells. The impact of the paper would benefit from the inclusion of references that relate directly to the role of IRF8 in antigen cross-presentation.

Related to this, it would be helpful for the authors to introduce what is known about IRF8 and macrophage development - since this may be more relevant to LC.

2. I am not an expert in the analysis of RNAseq/single cell RNAseq and ChIPseq data and found the text and figures difficult to follow and interpret as they are presented. Many abbreviations are not spelt out.

i/ Figure 2A - what do the numbers on the X axis mean? How does the overlap between antigen cross-presentation genes and migrated LC compare to DC1 and DC2 - is the overlap the same as for DC1? Is

there no overlap with DC2?

ii/ The rationale for performing the the single cell RNAseq analyses is not obvious and the discussion of the results comes across as being very descriptive. The authors identify 3 subsets/clusters and propose the subset A is programmed "for both tolerogenic CD4 and protective CD8 responses" - but there is no further data to support this hypothesis, nor evidence that these subsets are relevant biologically.

iii/The text for figure 6 is very long, and difficult to follow. There is no reference to supplemental figure 6 (I did not understand what purpose supp figure 6B served, and it is very poor resolution). What is the rationale for showing the very limited numbers of differentially expressed genes in figures 6A and B? Was this a subjective or objective selection of genes? Figure 6C includes H3K27Me DEGs but the concept of H3K27 methylation has not yet been introduced in the paper. Also the abbreviation "DEG" is not explained.

3. The paper relies heavily on the supplemental figures, but these are not well-constructed and inconsistent e.g. some flow plots have the numbers missing from their axes or the ticks, others are poor resolution.

4. The discussion is highly speculative and should be re-written. In particular the discussion on page 20-21 about cross-talk between LC and other cells is not relevant since this has not been addressed in the manuscript (this is also in the introduction but not studied). The authors again state "In cDC1, one of two main blood DC subsets, cross-presentation has been shown to depend critically on IRF8 and BATF interaction with composite elements on the promoters of key target genes". But this is not referenced.

Reviewer #2 (Langerhans/dendritic cell functions)(Remarks to the Author):

Polak et al. report that the ability of human LCs to cross-present correlates with the up-regulation of IRF4. The concept is important as it strengthens previous studies on the functional capacity of LCs to cross present, in addition it coincide with the ability of murine mo-DCs that express IRF4 to cross-present riseño et al. 2016, however, the paper lack significant rigor.

It is not clear why the authors selected TNF as the sole activator that was examined. CD40L is known to enhance cross-presentation and should be included.

More than one antigen should be tested in a cross-presentation assay.

Fig. 3 only 138 cells were analyzed this is extremely low, is the comparison to the different blood DCs significant?

Fig. 5a, the figure is not clear. Single cell PCR should be performed for key genes in the antigen cross-presentation pathway and the IRF4. Alternatively, proteins in the pathways should be stained in addition to IRF4

Figure 1b: gating strategy for the migrated and ss LCs is not clear. Seems different in the 2 culture conditions.

Figure 6 a: no scale is added to the level of expression—it's impossible to interpret this figure. The author should analyze the protein expression.

Figure 6D, the marks are not clear

The method for purifying the cells is not clear. Specifically, in the case of steady state, more details are needed. Also the purity with CD1a beads may not be as optimal as sorting. Same in the case of mouse LCs.

Fig, 6 the author discuss time course following TNF, can LCs efficiently cross present 2h post TNFa activation?

Supplementary Table S6 shows TNF α kinetics clusters but the text says that it is supposed to show comparison of Chip-Seq data of ss LCs with mono and macs. Not sure where Chip-Seq data for ss LCs.

Figure S4g-h no stat is presented or a positive control with a short peptide. As well as statistics of multiple experiments. Do murine LCs express IRF4 in ss and how is it changed upon migration?

Minor:

Briseño et al. 2016, should be discussed and cited.

In addition, the work supports pioneering work performed with in-vitro LCs (Klechevsky et al. 2008, should be discussed and cited.

Reviewer #3 (Transcriptomic/epigenetic regulation of haematopoiesis)(Remarks to the Author):

Genetic programming of antigen cross-presentation in IRF4-expressing human Langerhans cells

The central hypothesis of the work highlighted here is that human Langerhan cells (LCs), a unique immune sentinel residing in the epidermis, achieve epidermal homeostasis in a pre-inflammatory setting by activating an IRF4-dependent class of genes that are specifically involved in cross-presentation of exogenous antigens to resident and infiltrating CD8+ T cells. This transcriptomic profile is sustained in the activated LC during an inflammatory response that can be triggered by pro-inflammatory cytokines secreted by other cutaneous cell types. The authors ascribe to this model a mechanism whereby in conjunction with its partners from ETS and AP-1 transcription factor families, IRF4 stabilizes the DNA binding activity of bound PU.1 and BATF at genes harboring EICE and AICE binding motifs, respectively. Irrespective of activation however, it appears the epigenetic landscape of H3K27Ac and H3K4me3 histone modifications implicated in the activation of such genes is largely already in place in the pre-inflammatory setting, or at least in what the authors refer to as the "steady-state migrated" LC.

As stated in the discussion, two of the largest obstacles facing the authors in studying human LCs, are 1) the lack of a source for readily available human samples and 2) the unreliability of using mouse epidermal tissue as a tractable system for elucidating LC function in humans. Thus, with a limited number of human samples the authors use previously established protocols to isolate a pure population of migrated human LCs that are at times cultured in vitro. One of the more enduring take home points from this work is that LCs are very much unlike conventional dendritic cells in their ability to upregulate genes involved in antigen processing and antigen cross-presentation without a concomitant increase in genes involved in cytokine production and cytokine-receptor signaling. This indeed is a unique feature of LCs that is imparted via IRF4-dependent processes, and the idea of LCs being well-suited to promote tissue homeostasis hinges on the finding that IRF8 does not seem to play a role specifically in the activation of ubiquitin-dependent proteolytic pathways that underpin antigen cross-presentation in human LCs. Individual RNA-FISH data results for some of the ubiquitin-dependent proteolytic pathway genes should be included at a minimum here to validate the Seq data. Unfortunately, the experiments carried out in this report do not definitively prove that Irf4 is indeed solely responsible for the apparent "unique" ability of LCs to tease apart genetic programs that are normally mutually exclusive in dendritic cells. On balance the authors seem unable to drill down the functional significance of Irf4 upregulation in steady-state migrated human LCs as it pertains to cross-presentation of antigen pathways that are important for immunological outcomes. Are LCs promoting tissue homeostasis by upregulating Irf4, or are they poised to program adaptive immune responses because of the specific upregulation of Irf4? Or is it both?

Comments

In Figure 1a, TNF-alpha is administered for two or twenty-four hours, but this only really applies to Figure 1e, and it not clear from the manuscript body or figure legend which time point of TNF-alpha incubation is used for the experiment in Figure 1e.

In Figure 2b an overlap of 53 genes between the three different gene signatures (cross-presenting genes, antigen processing, and steady-state migrated LC genes) is shown, however in the body of text on page 7 it is summarized as "57 genes shared between all three subsets....". An explanation for this discrepancy should be noted.

On page 8, the sentence beginning "The coupling of oxidative phosphorylation...." is a sentence fragment. Please review the sentence construction here. In the same paragraph, the authors make the assertion that "LC subset A cells appear to be optimally programmed for priming of tolerogenic CD4 and protective CD8 T cell responses."

Another critique of the scRNAseq data is that it is composed of only the steady-state migrated LC condition, and does not include a TNF-alpha-stimulated cohort. There is also no mention of the significance of the other LC subsets beyond subset A. The authors should consider elaborating more on the functional significance of each subset, because it seems the audience stands to potentially learn a lot more from this division of subsets.

On page 10 the authors describe an experiment using murine IRF8-deficient LCs that cross-present antigen to CD8 cytotoxic T cells that are just as effective as their WT counterparts in eliciting a response. If it is the authors' goal to demonstrate a role for IRF4 in cross-presentation, then the better experiment to do is to use IRF4 KO mice. This is especially important considering the authors have stated that caution should be exercised when extrapolating insights from the mouse epidermis for human LC function.

In Figure 4d a FACS 2d plot for the two-hour time point of TNF-alpha incubation is not included. There is however an inflection point for IRF8 transcripts after 2 hours of TNF-alpha incubation so can the authors really state that IRF8 is never expressed during 24-hour TNF-alpha time course?

On page 15, the sentence beginning "three samples (Supplementary data...." Makes a mention of using steady-state LCs, but it seems authors mean steady-state "migrated" cells were used. It's a confusing point

On page 16 authors make inconsistent statements regarding the upregulation of H3K27ac marks in genes underpinning LC activation. Moreover, it is not clear from any of the panels from Figure 6a-d that this upregulation is significant, and the authors themselves state that these marks are readily detected prior to activation.

On page 17 the authors make the statement, "Thus, transcription and chromatin profiling coupled with TF motif analysis strong suggests....". The word "strong" in this sentence should be changed to "strongly".

On page 20 the sentence that begins, "Since abrogation of polyubiquitination strongly reduces...." Is a sentence fragment ending with the word "activated."

Point-by-point responses to reviewers:

Reviewer #1 (Skin APC functions)(Remarks to the Author):

Langerhans cells (LC) were originally defined as prototypic DC, and the processes of maturation and migration to draining LN for stimulation of naive CD4 and CD8 T cells considered a paradigm for DC behaviour in other tissues. However, our more recent understanding that LC are derived from macrophage lineage cells has led the field to question our understanding of LC function. Here the authors investigate the proteins that programme antigen presentation in LC under the control of the transcription factors IRF4 and 8 that are important for the development, and potentially function, of DC1 and DC2 respectively. The key finding is that in contrast to cDC1, IRF4, not IRF8, is important for the activation of genes associated with antigen processing and cross-presentation in LC and this process is initiated as LC migrate out of the epidermis.

This is a comprehensive study in which the authors have used a range of tools with which to directly investigate the factors controlling antigen presentation in human LC. I have some concerns however that should be addressed:

1. The study relies heavily on interpretation of transcriptomic and ChIPseq data to infer conclusions about the links between IRF4 and antigen cross-presentation. There is some mechanistic data using the IRF4^{-/-} mouse, but these experiments are very limited and shown in the supplemental data (see also my comment 2.). The paper would benefit from experiments such as those used by the group for their 2014 Nature Immunology paper (DOI: 10.1038/ni.2795) - e.g What is the impact of the conditional deletion of IRF4 in LC? Do these cells still present and cross-present antigen in vitro and topically applied antigen in vivo? What is the effect on formation of peptide-MHC I/II complexes in IRF4^{-/-} LC?

We thank the reviewer for assessing our study as comprehensive and interesting, noting in particular our ability to examine the genomic programming of **primary human LCs** and to couple this analysis with their epigenomic, cellular and functional properties. We agree with the reviewer that a major shortcoming of the earlier version of the manuscript was the absence of experiments examining the effects of perturbing IRF4 in human LCs. To address this criticism, we have directly tested the role of IRF4 in controlling the transcriptomic programme of human LCs (Fig. 6, Fig. S6 and Tables 9, 10). Importantly this involved CRISPR-Cas9 based editing of the *IRF4* gene in primary LCs, and analysing the changes in gene expression using scRNAseq. We confirmed that the editing of the *IRF4* gene resulted in not only knock down of *IRF4* transcripts but crucially, also the protein. These experiments revealed that IRF4 positively regulates the LC activation program including expression of ubiquitin pathway components involved in antigen processing while repressing NF2EL2 and NFkB pathway genes that promote responsiveness to oxidative stress and inflammatory cytokines. Importantly, these new results suggest that IRF4-dependent genomic programming of human migratory LCs enables their maturation while attenuating excessive inflammatory and immunogenic responses in the epidermis thereby promoting homeostasis. Since IRF4 knockdown LCs begin to manifest survival defects after 48h in culture (time required for genome editing) we utilized scRNAseq analysis for analysing their mutant

phenotype. Unfortunately, we were not able to use IRF4 knockdown LCs for various functional analyses involving antigen presentation and priming of T cell responses. Nevertheless, our experiments conclusively demonstrate that IRF4 is indeed a key orchestrator of human LC genomic programming and therefore function.

2. IFN γ production by recently primed OT-I cells is a poor readout for in vitro assays with DC/LC (Supp figure 4), and as a result the data are not robust. T cell division (e.g. CFSE dilution) would be a clearer read-out (again as used in the 2014 paper) with down-regulation of CD62L or up-regulation of CD25/CD69. It would be helpful to compare wild-type and IRF8-deficient DC1 in these assays.

We thank the reviewer for this comment. We have previously demonstrated expansion of antigen-specific CD8 T cells induced by LCs in a proliferation assay, in our previously published research article in Journal of Investigative Dermatology, 2012, (1). Given the tighter focus of the revised manuscript on the functions of IRF4 in the genomic programming of human LCs including the new genome editing experiments that directly test the requirement of IRF4, we have removed the supplementary data focusing on IRF8 deficient murine LCs in their priming of OT-I responses.

I also found the paper difficult to read, and suggest that the authors consider re-writing it if it were to be published in a broad science paper such as Nature Communications. For example:

1. The authors frequently allude to a fundamental role for IRF8 in cross-presentation by DC and this assertion is key to their conclusion that LC cross-present antigen despite an absence of IRF8. However, I am unaware of work showing a direct role for IRF8 for the cross-presentation machinery in DC1. A number of references are cited in the introduction, but they all refer to the interaction between IRF4 and presentation on MHCII to CD4 T cells. The impact of the paper would benefit from the inclusion of references that relate directly to the role of IRF8 in antigen cross-presentation.

Related to this, it would be helpful for the authors to introduce what is known about IRF8 and macrophage development - since this may be more relevant to LC.

We have re-written parts of the manuscript to make it more comprehensible to a wider audience. We have also included the relevant citations concerning the role of IRF8 in cross-presentation. In our introduction, we state that "Furthermore, in murine CD8a DCs cross-presentation is critically dependent on BATF3/IRF8 complexes⁴⁴⁻⁴⁷" p4, line 118-119.

2. I am not an expert in the analysis of RNAseq/single cell RNAseq and ChIPseq data and found the text and figures difficult to follow and interpret as they are presented. Many abbreviations are not spelt out.

The list of abbreviations has been added.

i/ Figure 2A - what do the numbers on the X axis mean? How does the overlap between antigen cross-presentation genes and migrated LC compare to DC1 and DC2 - is the overlap the same as for DC1? Is there no overlap with DC2?

The x-axis shows consecutive cut-offs for each interval in gene expression levels. This has now been added to the legend.

ii/ The rationale for performing the the single cell RNAseq analyses is not obvious and the discussion of the results comes across as being very descriptive. The authors identify 3 subsets/clusters and propose the subset A is programmed “for both tolerogenic CD4 and protective CD8 responses” - but there is no further data to support this hypothesis, nor evidence that these subsets are relevant biologically.

We have substantially expanded our analysis of the scRNAseq analyses of human LCs, including new figure panels with gene annotated bar plots that facilitate understanding of the key molecular signatures distinguishing the various cell clusters (Figure 3). The increased numbers of cells analysed confirm our earlier findings and the pseudo trajectory analysis reveals that the programme for LC activation and antigen presentation evolves across the LC clusters #3→#1→#2. We agree that the interpretations of the functions of these gene expression modules is to some degree hypothetical, nevertheless they are well-grounded in the wealth of studies linking coordinated gene expression, i.e. transcriptomic programmes, with cell biology and function (2,3). We have worded the text cautiously, to make sure we do not over interpret the data.

iii/The text for figure 6 is very long, and difficult to follow. There is no reference to supplemental figure 6 (I did not understand what purpose supp figure 6B served, and it is very poor resolution). What is the rationale for showing the very limited numbers of differentially expressed genes in figures 6A and B? Was this a subjective or objective selection of genes? Figure 6C includes H3K27Me DEGs but the concept of H3K27 methylation has not yet been introduced in the paper. Also the abbreviation “DEG” is not explained.

We apologize for the difficulties in interpreting this figure and the accompanying text. With the new CRISPR-Cas9 editing experiments, the old figures 5 and 6 either have been re-placed or re-arranged. Figure 5 is superseded by the CRISPR-Cas9 editing experiments, and data presented in Figure 6 split for RNA-seq data (Fig. 2) and separately for chromatin changes (Fig. 4). We hope that the new figure order and the accompanying text makes the logical flow of the manuscript easier to follow.

3. The paper relies heavily on the supplemental figures, but these are not well-constructed and inconsistent e.g. some flow plots have the numbers missing from their axes or the ticks, others are poor resolution.

We apologise for the poor quality of the supplementary figures. We believe the low resolution was caused by conversion of original files to pdf. We have improved the quality of the supplementary figures in this revised submission.

4. The discussion is highly speculative and should be re-written. In particular the discussion on page 20-21 about cross-talk between LC and other cells is not relevant since this has not been addressed in the manuscript (this is also in the introduction but not studied). The authors again state “In cDC1, one of two main blood DC subsets, cross-presentation has

been shown to depend critically on IRF8 and BATF interaction with composite elements on the promoters of key target genes". But this is not referenced.

We have revised the discussion to strike a better balance by focusing primarily on our findings and their implications as well as relationships with other insights in the literature.

Reviewer #2 (Langerhans/dendritic cell functions)(Remarks to the Author):

Polak et al. report that the ability of human LCs to cross-present correlates with the up-regulation of IRF4. The concept is important as it strengthens previous studies on the functional capacity of LCs to cross present, in addition it coincides with the ability of murine mo-DCs that express IRF4 to cross-present riseño et al. 2016, however, the paper lacks significant rigor. It is not clear why the authors selected TNF as the sole activator that was examined. CD40L is known to enhance cross-presentation and should be included. More than one antigen should be tested in a cross-presentation assay.

We thank the reviewer for stating that one of the key concepts underpinning the manuscript is important. While we agree that TNF α is not the only activator of LCs, we chose it since it is relevant to skin inflammation. We have previously documented, that CD40 stimulation can induce both CD70 expression and ability of LCs to cross-present (4). While it would be of interest to pursue studies of this interaction, and how transcriptomic programmes are regulated by CD40 signalling in human LCs we have substantially expanded the current work by a more extensive scRNAseq analysis of migratory human LCs (Fig. 3) and CRISPR-Cas9 based genome editing of IRF4 (Fig. 6). We feel that our analysis can provide the foundational framework for future studies of the genomic responses of human LCs to distinct stimuli.

Fig. 3 only 138 cells were analyzed this is extremely low, is the comparison to the different blood DCs significant?

We have now expanded the number of scRNA data points from 138 to above 2000 (including 950 migrated LCs and 1484 cells in CRISPR-Cas9 editing experiment), thereby validating the previous clusters and the enriched gene modules (Fig. 3). We have also repeated the projection of the LC transcriptome to 6 other DCs type described by Villani et al (2) and confirmed the resemblance of LCs to DC2 and DC3, but not DC1 (New data is in the Supplementary figure S3e).

Fig. 5a, the figure is not clear. Single cell PCR should be performed for key genes in the antigen cross-presentation pathway and the IRF4. Alternatively, proteins in the pathways should be stained in addition to IRF4

We thank the reviewer for pointing out this shortcoming. Expression of key proteins important for antigen processing was confirmed by FACS, and added to the main manuscript, Figure 2c

Figure 1b: gating strategy for the migrated and ss LCs is not clear. Seems different in the 2 culture conditions.

LCs undergo substantial changes in their morphology accompanied with major alterations in protein expression (e.g., CD1a, CD207, HLADR) during their migration from the epidermis. The gating strategy was adjusted to capture LCs in these two different states.

Figure 6 a: no scale is added to the level of expression—it's impossible to interpret this figure. The author should analyze the protein expression.

We apologise for the omission. The figure is now included as Figure 2d, the scale has been added accordingly. The expression of proteins encoded by key genes involved in LC activation and antigen processing (CD40, CD83, SQSTM1 and TRIM21, Fig. 1c, Figure 2c, Supplementary Figures S1a and S2g) has been assessed by flow cytometry.

Figure 6D, the marks are not clear

The method for purifying the cells is not clear. Specifically, in the case of steady state, more details are needed. Also the purity with CD1a beads may not be as optimal as sorting. Same in the case of mouse LCs.

LCs were migrated for 48h or longer. The purity of LCs used for RNA-seq and ChIP-seq was confirmed post-magnetic bead isolation, and found consistently to be around 95%. Our extensive scRNAseq analyses independently confirmed the purity of our LC preparations (Fig. 3). While flow sorting can potentially yield even purer populations, the shear stresses encountered during sorting, affects LC viability and the recovered RNA yield. Additionally, the time required for sorting can significantly delay the cell isolation for genomic and epigenomic analyses.

Fig, 6 the author discuss time course following TNF, can LCs efficiently cross present 2h post TNFa activation?

This is a very interesting question, as LCs are able to cross-present directly after migration from epidermis. Thus most likely they are also able to cross-present after 2h stimulation with TNFa, but we have not performed the experiment.

Supplementary Table S6 shows TNFa kinetics clusters but the text says that it is supposed to show comparison of Chip-Seq data of ss LCs with mono and macs. Not sure where Chip-Seq data for ss LCs.

We apologise for the confusion. The data in Supplementary Table S6 shows enrichment scores Z for migrated LCs at each time point. The time point referenced in the text was T0, representing unstimulated migrated LCs. This has now been corrected.

Figure S4g-h no stat is presented or a positive control with a short peptide. As well as statistics of multiple experiments. Do murine LCs express IRF4 in ss and how is it changed upon migration?

The experiment with murine LCs involved pooled LCs from 3 mice. Given the tighter focus of the revised manuscript on the functions of IRF4 in the genomic programming of human LCs

including the new genome editing experiments that directly test the requirement of IRF4, we have removed the supplementary data focusing on IRF8 deficient murine LCs in their priming of OT-I responses.

Minor:

Briseño et al. 2016, should be discussed and cited.

In addition, the work supports pioneering work performed with in-vitro LCs (Klechevsky et al. 2008, should be discussed and cited.

Briseño et al. 2016 (5) is now cited and discussed on p21, lines 538-542: “By contrast, recent reports in other cell types, such as MoDCs⁷⁵, cDC2 (K.Murphy, personal communication) indicate that the same transcriptomic programme can be successfully initiated and executed by IRF4⁷⁵. Hence the high levels of IRF4 expressed in human migrated LCs, are likely to be involved in the orchestration of this programme⁷⁵”.

Klechevsky et al 2008 (6) is now cited on p3 line 78 “LCs have also been shown to be capable of efficient cross-presentation in which exogenous antigens are presented on MHC class I, resulting in activation and expansion of antigen-specific effector CD8 T cells^{2,5,9,10}” and p6 lines 145 and 151, as the key paper describing work using *in-vitro* LCs: “In agreement with our earlier findings and those of others^{2,9,10}, human LC after migrating from the epidermis underwent maturation and uniformly expressed high levels of CD1a, CD207 and HLA-DR” and “Such migrated LCs have been shown to efficiently present antigens to CD4 as well as CD8 T cells^{2,5,9,10,24,25}”.

Reviewer #3 (Transcriptomic/epigenetic regulation of haematopoiesis)(Remarks to the Author):

Genetic programming of antigen cross-presentation in IRF4-expressing human Langerhans cells

The central hypothesis of the work highlighted here is that human Langerhan cells (LCs), a unique immune sentinel residing in the epidermis, achieve epidermal homeostasis in a pre-inflammatory setting by activating an IRF4-dependent class of genes that are specifically involved in cross-presentation of exogenous antigens to resident and infiltrating CD8+ T cells. This transcriptomic profile is sustained in the activated LC during an inflammatory response that can be triggered by pro-inflammatory cytokines secreted by other cutaneous cell types. The authors ascribe to this model a mechanism whereby in conjunction with its partners from ETS and AP-1 transcription factor families, IRF4 stabilizes the DNA binding activity of bound PU.1 and BATF at genes harboring EICE and AICE binding motifs, respectively. Irrespective of activation however, it appears the epigenetic landscape of H3K27Ac and H3K4me3 histone modifications implicated in the activation of such genes is largely already in place in the pre-inflammatory setting, or at least in what the authors refer to as the “steady-state migrated” LC.

As stated in the discussion, two of the largest obstacles facing the authors in studying human LCs, are 1) the lack of a source for readily available human samples and 2) the

unreliability of using mouse epidermal tissue as a tractable system for elucidating LC function in humans. Thus, with a limited number of human samples the authors use previously established protocols to isolate a pure population of migrated human LCs that are at times cultured in vitro. One of the more enduring take home points from this work is that LCs are very much unlike conventional dendritic cells in their ability to upregulate genes involved in antigen processing and antigen cross-presentation without a concomitant increase in genes involved in cytokine production and cytokine-receptor signaling. This indeed is a unique feature of LCs that is imparted via IRF4-dependent processes, and the idea of LCs being well-suited to promote tissue homeostasis hinges on the finding that IRF8 does not seem to play a role specifically in the activation of ubiquitin-dependent proteolytic pathways that underpin antigen cross-presentation in human LCs. Individual RNA-FISH data results for some of the ubiquitin-dependent proteolytic pathway genes should be included at a minimum here to validate the Seq data. Unfortunately, the experiments carried out in this report do not definitively prove that Irf4 is indeed solely responsible for the apparent “unique” ability of LCs to tease apart genetic programs that are normally mutually exclusive in dendritic cells. On balance the authors seem unable to drill down the functional significance of Irf4 upregulation in steady-state migrated human LCs as it pertains to cross-presentation of antigen pathways that are important for immunological outcomes. Are LCs promoting tissue homeostasis by upregulating Irf4, or are they poised to program adaptive immune responses because of the specific upregulation of Irf4? Or is it both?

We thank the reviewer for their in-depth review of our manuscript and in raising the central issue. We agree with the reviewer that a major shortcoming of the earlier version of the manuscript was the absence of experiments examining the effects of perturbing IRF4 in human LCs. To address this criticism, we have directly tested the role of IRF4 in controlling the transcriptomic programme of human LCs (Fig. 6, Fig. S6 and Tables 9, 10). Importantly this involved CRISPR-Cas9 based editing of the *IRF4* gene in primary LCs, and analysing the changes in gene expression using scRNAseq. We confirmed that editing of the *IRF4* gene resulted in not only knock down of *IRF4* transcripts but crucially also the protein. These experiments revealed that IRF4 positively regulates the LC activation program including expression of ubiquitin pathway components involved in antigen processing while repressing NF2EL2 and NFkB pathway genes that promote responsiveness to oxidative stress and inflammatory cytokines. Importantly, these new results suggest that IRF4-dependent genomic programming of human migratory LCs enables their maturation while attenuating excessive inflammatory and immunogenic responses in the epidermis thereby promoting homeostasis.

Comments

In Figure 1a, TNF-alpha is administered for two or twenty-four hours, but this only really applies to Figure 1e, and it not clear from the manuscript body or figure legend which time point of TNF-alpha incubation is used for the experiment in Figure 1e.

It is 24h, we apologise for the lack of clarity. The detail has been added to the figure legend. *In Figure 2b an overlap of 53 genes between the three different gene signatures (cross-presenting genes, antigen processing, and steady-state migrated LC genes) is shown, however in the body of text on page 7 it is summarized as “57 genes shared between all three subsets....”. An explanation for this discrepancy should be noted.*

We apologise for this typographical mistake, the number of genes has now been adjusted to the correct value consistently across the text and figure.

On page 8, the sentence beginning “The coupling of oxidative phosphorylation....” is a sentence fragment. Please review the sentence construction here. In the same paragraph, the authors make the assertion that “LC subset A cells appear to be optimally programmed for priming of tolerogenic CD4 and protective CD8 T cell responses.”

We thank the reviewer for identifying this mistake, the text of the manuscript has now been edited, and we hope that all the linguistic shortcomings were corrected.

Another critique of the scRNAseq data is that it is composed of only the steady-state migrated LC condition, and does not include a TNF-alpha-stimulated cohort. There is also no mention of the significance of the other LC subsets beyond subset A. The authors should consider elaborating more on the functional significance of each subset, because it seems the audience stands to potentially learn a lot more from this division of subsets.

We have substantially expanded our analysis of the scRNAseq analyses of human LCs, including new figure panels with gene annotated bar plots that facilitate understanding of the key molecular signatures distinguishing the various cell clusters (Figure 3). The number of scRNA data points have been expanded from 138 to above 2000 (including 950 migrated LCs), validating the previous clusters and the enriched gene modules (Fig. 3). We have also repeated the projection of the LC transcriptome to 6 other DCs type described by Villani et al (2), and confirmed the resemblance of LCs to DC2 and DC3, but not DC1 (New data is in the Supplementary figure S3e). The increased numbers of cells analysed confirm our earlier findings of 3 major clusters and the pseudo trajectory analysis reveals that the programme for LC activation and antigen presentation evolves across the LC clusters in the following order #3→#1→#2. Although the interpretations of the functions of these gene expression modules are to some degree hypothetical, nevertheless they are well-grounded in the wealth of studies linking coordinated gene expression, i.e. transcriptomic programmes, with cell biology and function (2,3). Given that our study identified the migration from the epidermis as the most pivotal event for genomic programming of LCs encompassing their maturation for antigen presentation, responsiveness to cytokines and the priming of T cell responses, we have focused our extensive scRNA analyses on migrated LCs. We feel that our scRNAseq analysis will provide the foundational framework for future studies of the genomic responses of human LCs to distinct stimuli including TNF α .

On page 10 the authors describe an experiment using murine IRF8-deficient LCs that cross-present antigen to CD8 cytotoxic T cells that are just as effective as their WT counterparts in eliciting a response. If it is the authors' goal to demonstrate a role for IRF4 in cross-presentation, then the better experiment to do is to use IRF4 KO mice. This is especially important considering the authors have stated that caution should be exercised when extrapolating insights from the mouse epidermis for human LC function.

We agree with the reviewer that the key experiment is the perturbation of IRF4 in Langerhans cells, particularly human LCs. As noted above, we have now directly tested the role of IRF4 in

controlling the transcriptomic programme of human LCs (Fig. 6, Fig. S6 and Tables 9, 10). Importantly this involved CRISPR-Cas9 based editing of the *IRF4* gene in primary LCs, and analysing the changes in gene expression using scRNAseq. Given the tighter focus of the revised manuscript on the functions of IRF4 in the genomic programming of human LCs including the new genome editing experiments that directly test the requirement of IRF4, we have removed the supplementary data focusing on IRF8 deficient murine LCs in their priming of OT-I responses.

In Figure 4d a FACS 2d plot for the two-hour time point of TNF-alpha incubation is not included. There is however an inflection point for IRF8 transcripts after 2 hours of TNF-alpha incubation so can the authors really state that IRF8 is never expressed during 24-hour TNF-alpha time course?

We thank the reviewer for this observation. The full course of IRF8 protein expression assessed by flow cytometry is shown in Supplementary Figure S5f. Importantly, despite the transient peak of mRNA expression at 2h (low levels of transcripts), the IRF8 protein is not expressed at detectable levels at this time point.

On page 15, the sentence beginning "three samples (Supplementary data...." Makes a mention of using steady-state LCs, but it seems authors mean steady-state "migrated" cells were used. It's a confusing point

We apologize for this confusion in the text. It is now corrected by removing the phrase "steady-state".

On page 16 authors make inconsistent statements regarding the upregulation of H3K27ac marks in genes underpinning LC activation. Moreover, it is not clear from any of the panels from Figure 6a-d that this upregulation is significant, and the authors themselves state that these marks are readily detected prior to activation.

We apologize for not being sufficiently clear. While TNFa does induce some enhancement in H3K27Ac histone marks around gene promoters and enhancers, especially in genes from the late phase response, we observe that the majority of gene regulatory sequences are marked by activating histone modifications in the unstimulated migrated LCs. We have altered the text accordingly.

On page 17 the authors make the statement, "Thus, transcription and chromatin profiling coupled with TF motif analysis strong suggests....". The word "strong" in this sentence should be changed to "strongly".

We have changed the word accordingly.

On page 20 the sentence that begins, "Since abrogation of polyubiquitination strongly reduces...." Is a sentence fragment ending with the word "activated."

This has now been edited and corrected

We thank the reviewers for their critical and thorough reviews of the manuscript and in challenging us to perform the key new experiments to substantiate and extend our main findings. We also appreciate the suggestions for improving the presentations of the data as well as the clarity of the text. The revised manuscript has been substantially improved as direct consequence of this invaluable input.

References

- 1) Polak, M. E. *et al.* CD70-CD27 interaction augments CD8+ T-cell activation by human epidermal Langerhans cells. *J Invest Dermatol* **132**, 1636-1644, doi:10.1038/jid.2012.26 (2012).
- 2) Villani, A. C. *et al.* Single-cell RNA-seq reveals new types of human blood dendritic cells, monocytes, and progenitors. *Science* **356**, doi:10.1126/science.aah4573 (2017).
- 3) Vander Lugt, B. *et al.* Transcriptional determinants of tolerogenic and immunogenic states during dendritic cell maturation. *J Cell Biol* **216**, 779-792, doi:10.1083/jcb.201512012 (2017).
- 4) White, A.L. *et al.* Conformation of the Human Immunoglobulin G2 Hinge Imparts Superagonistic Properties to Immunostimulatory Anticancer Antibodies. *Cancer Cell* (2015) 27(1):138-48.
- 5) Briseno, C. G. *et al.* Distinct Transcriptional Programs Control Cross-Priming in Classical and Monocyte-Derived Dendritic Cells. *Cell Rep* **15**, 2462-2474, doi:10.1016/j.celrep.2016.05.025 (2016).
- 6) Klechevsky, E. *et al.* Functional specializations of human epidermal Langerhans cells and CD14+ dermal dendritic cells. *Immunity* **29**, 497-510, doi:10.1016/j.immuni.2008.07.013 (2008).

REVIEWERS' COMMENTS:

Reviewer #1 (Remarks to the Author):

The authors have clearly put in a huge amount of effort to revise their manuscript based on the the reviewers' comments. The result is a significantly approved piece of work, in which the authors have clearly taken on board the suggestions made by the reviewers. The manuscript is also vastly improved in its readability and the quality of the figures. The result is an excellent and highly significant study that will be of great interest to the LC/macrophage/DC communities.

I have 2 very minor comments:

1. For the new figure 6a and b. If the summary graph in b is taken from histograms such as shown in a, then it would be more appropriate to plot the MFI for IRF4, since there is not a distinct + or - population. Is this is not the case then the authors should show the flow plots used to determine % IRF4+ LC.
2. The discussion is much better but still very long. I would suggest removing the section on "trained" cells (lines 521-527), which is highly speculative since the authors do not know the epigenetic status of LC at birth. I would also remove most of the last paragraph, which is also mainly speculation.

Reviewer #2 (Remarks to the Author):

Although there is a substantial improvement to the manuscript, the key experiment linking IRF4 to cross presentation in LCs has not been done. now that the authors were successful in deleting IRF4 in primary human LCs, and have removed all the mouse data, they should perform the key experiment of cross-presentation using these IRF4 modified cells, alternatively they can show that any of the IRF4 modulated genes impact LCs function.

also it doesn't look like an additional antigen was added to the crossp assay in spite of our request. even if this was not done , the author should explain better what T cells were used ? was it a cell line? were the CD8 purified or this is just PBMCs? and if so what is the % of the specific cells? non of the details are clear

Reviewer #3 (Remarks to the Author):

The manuscript is much improved. It is now suitable for publication.

Point by point response to reviewers

Reviewer #1 (Remarks to the Author):

The authors have clearly put in a huge amount of effort to revise their manuscript based on the reviewers' comments. The result is a significantly improved piece of work, in which the authors have clearly taken on board the suggestions made by the reviewers. The manuscript is also vastly improved in its readability and the quality of the figures. The result is an excellent and highly significant study that will be of great interest to the LC/macrophage/DC communities.

I have 2 very minor comments:

- 1. For the new figure 6a and b. If the summary graph in b is taken from histograms such as shown in a, then it would be more appropriate to plot the MFI for IRF4, since there is not a distinct + or - population. If this is not the case then the authors should show the flow plots used to determine % IRF4+ LC.*
- 2. The discussion is much better but still very long. I would suggest removing the section on "trained" cells (lines 521-527), which is highly speculative since the authors do not know the epigenetic status of LC at birth. I would also remove most of the last paragraph, which is also mainly speculation.*

We thank the reviewer for finding our revised manuscript to be significantly improved and for stating that this study will be of great interest to the LC/macrophage/DC communities.

1. We have replaced the plot comparing the percentage contribution of IRF4+ LCs with a plot summarising MFI values across the replicate experiments (Figure 6a).
2. We have revised the discussion by removing the paragraph on trained immunity and also shortening the last paragraph and making it less speculative.

Reviewer #2 (Remarks to the Author):

Although there is a substantial improvement to the manuscript, the key experiment linking IRF4 to cross presentation in LCs has not been done. Now that the authors were successful in deleting IRF4 in primary human LCs, and have removed all the mouse data, they should perform the key experiment of cross-presentation using these IRF4 modified cells, alternatively they can show that any of the IRF4 modulated genes impact LCs function.

We thank the reviewer for recognising the substantial improvement of the revised manuscript. We agree, that the suggested functional experiments using IRF4 edited primary LCs would be an excellent addition. Indeed, after having established a protocol for Cas9 mediated genome editing of human LCs we considered such an experiment. However, this was not technically feasible for the following reasons (i) short lifespan of primary human LCs in culture (ii) limiting numbers of cells and (iii) genome editing occurs in a fraction of the cells and it is not possible to sort cells with reduced expression of IRF4 as this requires cellular fixation and permeabilization. Given, these experimental constraints we decided to use an

experimental design that involved single cell analysis and importantly could distinguish IRF4 expressing LCs from their IRF4 deleted counterparts. Thus, scRNA-seq analysis of the heterogenous IRF4 edited population of LCs enabled us to successfully examine changes in transcriptional modules and genes that are regulated by IRF4. Using this powerful genomic methodology, we were able to demonstrate that genes encoding key components of myeloid cell activation as well as the ubiquitination pathway (inferred from the epigenetic analyses) were regulated by IRF4 (Figure 6d). Crucially, the unbiased analysis of control and IRF4 knockdown LCs enabled us to uncover novel functions of IRF4 in human LCs in a broader context. To reflect the broader relevance of IRF4 in human LCs the manuscript title was changed to “Genomic Programming of IRF4-expressing human Langerhans cells”.

also it doesn't look like an additional antigen was added to the crossp assay in spite of our request. even if this was not done , the author should explain better what T cells were used ? was it a cell line? were the CD8 purified or this is just PBMCs? and if so what is the % of the specific cells? non of the details are clear

The T cells used in the experiments were GLC-specific cell lines from HLA-A2 positive donors (Polak et al 2012). We apologise for the lack of clarity. A section explaining how CD8 T cell lines were generated has been expanded in the methods section, lines 469:474

Reviewer #3 (Remarks to the Author):

The manuscript is much improved. It is now suitable for publication.

We thank the reviewer for finding our manuscript to be substantially improved and endorsing its publication.